



# Calibration and evaluation of broad supersaturation scanning (BS2) cloud condensation nuclei counter for rapid measurement of particle hygroscopicity and CCN activity

Najin Kim[1], Yafang Cheng[2], Nan Ma[3], Mira L. Pöhlker[1], Thomas Klimach[1], Thomas. F. Mentel[4], Ovid O. Krüger [1], Ulrich Pöschl[1] and Hang Su[1*]

[1]Multiphase Chemistry Department, Max Planck Institute for Chemistry, Mainz, 55128, Germany
[2]Minerva Research Group, Max Planck Institute for Chemistry, Mainz, 55128, Germany
[3]Center for Air Pollution and Climate Change Research (APCC), Institute for Environmental and Climate Research (ECI), Jinan University, Guangzhou, 511443, China
[4]Institute of Energy and Climate Research, IEK-8: Troposhere, Forschungzentrum Jülich GmbH, Jülich, 52425, Germany

*Correspondence to*: Dr. Hang Su (h.su@mpic.de)

**Abstract.** For understanding and assessing aerosol-cloud interactions and their impact on climate, reliable measurement data of aerosol particle hygroscopicity and cloud condensation nuclei (CCN) activity are required. The CCN activity of aerosol particles can be determined by scanning particle size and supersaturation ($S$) in CCN measurements. Compared to the existing differential mobility analyzer (DMA)-CCN activity measurement, a broad supersaturation scanning CCN (BS2-CCN) system, in which particles are exposed to a range of $S$ simultaneously, can measure the CCN activity with a high time-resolution. Based on a monotonic relation between the activation supersaturation of aerosol particles ($S_{aerosol}$) and the activated fraction ($F_{act}$) of the BS2-CCN measurement, we can derive κ, a single hygroscopicity parameter, directly. Here, we describe how the BS2-CCN system can be effectively calibrated and which factors can affect the calibration curve ($F_{act} - S_{aerosol}$). For calibration, size-resolved CCN measurements with ammonium sulfate and sodium chloride particles are performed under the three different thermal gradient (dT) conditions (dT=6, 8, and 10 K). We point out key processes that can affect the calibration curve and thereby need to be considered as follows: First, the shape of the calibration curve is primarily influenced by $S_{max}$, the maximum $S$ in the activation tube. We need to determine appropriate $S_{max}$ depending on particle size and κ to be investigated. To minimize the effect of multiply charged particles, small geometric mean diameter ($D_g$) and $\sigma_g$ geometric standard deviation ($\sigma_g$) in number size distribution are recommended when generating the calibration aerosols. Last, $F_{act}$ is affected by particle number concentration and has a decreasing rate of $0.02/100 cm^{-3}$ due to the water consumption in the activation tube. For evaluating the BS2-CCN system, inter-comparison experiments between typical DMA-CCN and BS2-CCN measurement were performed with the laboratory-generated aerosol mixture and ambient aerosols. Good agreements of κ values between DMA-CCN and BS2-CCN measurements for both experiments show that the BS2-CCN system can measure CCN activity well compared to the existing measurement, and can measure a broad range of hygroscopicity distribution with a high time-



resolution (~1 second vs. few minutes for a standard CCN activity measurement). As the hygroscopicity can be used as a proxy for the chemical composition, our method can also serve as a complementary approach for fast and size-resolved
detection/estimation of aerosol chemical composition.

## 1 Introduction

Atmospheric aerosol particles affect global climate change in that aerosols alter the radiative balance by scattering and absorbing shortwave and longwave radiation directly. Additionally, by serving as cloud condensation nuclei (CCN), atmospheric aerosol particles influence the radiative budget by modulating the microphysical structure, lifetime, and coverage
of the clouds. Although the direct and indirect effects of aerosols on climate change are widely accepted within the scientific community, the assessment of aerosol-cloud interactions and the quantification of their effect on climate still remain largely uncertain (IPCC 2013). Notably, one of the underlying challenges is to determine the ability of aerosol particles that act as cloud droplet, CCN activity, which has come up as a rising issue over the past years (McFiggans et al., 2006; Andreae and Rosenfeld, 2008; Hiranuma et al., 2011; Paramonov et al., 2013).
The CCN activity, the fraction at which aerosol particles can be activated to become CCN, can be determined by particle size and chemical composition at a given water vapor supersaturation (Charlson et al., 2001; Andreae et al., 2005, 2007; McFiggans et al., 2006; Cai et al., 2018) and can be parameterized by a single parameter, κ (Petters and Kreidenweis, 2007). The activation $S$ of aerosol particles can be estimated from dry particle diameter ($D_d$) and water activity (Kohler, 1936; Su et al., 2010). Once determined, the parameter κ can greatly simplify the descriptions of chemical composition effect in models.
κ is closely linked to the chemical compositions of aerosol particles, especially the ratio of organic to inorganic compositions. Thus, κ is expected to and has been demonstrated to show a size dependence due to the change of chemical compositions at different size ranges. However, size-resolved κ data are still limited due to the relatively slow response of the commercial instrument when scanning both $S$ and $D_d$ (Robert and Nenes 2005; Dusek et al., 2006; Moore and Nenes 2009a; Petters et al., 2009; Svenningsson et al., 2005; Wex et al., 2009; Rose et al., 2011; Zhao et al., 2015). For example, measurements based on
the continuous-flow streamwise thermal-gradient CCN-Counter (CCNC) from Droplet Measurement Technologies (DMT) complete a cycle of size-resolved κ in about an hour due to the slow temperature stabilization (about few minutes to stabilize) for changing new $S$. To solve the low-time resolution problem, other techniques have been developed such as "Scanning Flow CCN Analysis" (SFCA) that changes flow rate over time in the growth chamber under the constant temperature gradient (Moore and Nenes, 2009) and an instantaneous CCN spectrometer with more than 40 channels of supersaturation resolution
from 0.01% to 1.0% (Hudson, 1989).
Su et al. (2016) introduce a new concept for the design of CCN counter, the broad supersaturation scanning (BS2) approach, for the rapid measurement of particle hygroscopicity and CCN activity. Compared to the measurement system of DMT-CCNC that applies a single $S$ at the centerline, aerosol particles of the BS2-CCN system are introduced with a wider inlet at a low sheath-to-aerosol flow ratio (SAR) and are exposed to a range of $S$ simultaneously. Through this new design of CCNC, we can



obtain a monotonic relation between the activated fraction of aerosols ($F_{act}$) and critical activation supersaturation ($S_{aerosol}$)

and thereby calculate the size-resolved $S_{aerosol}$ as well as κ directly. A decrease in time required for scanning $S$ of BS2-CCN

system makes it possible to derive the κ with a high temporal and size resolution. Additionally, a constant temperature

difference minimizes the bias from the potential volatilization of aerosols in the instrument (Moore and Nenes, 2009).

In this study, we introduce the experimental setup of materialized BS2-CCN system, including the newly designed inlet, and

describe how the BS2-CCN system can be effectively calibrated. To validate the performance of the BS2-CCN system, we

perform the inter-comparison with the existing DMA-CCN measurement using the laboratory-generated aerosol mixture and

ambient aerosols.

## 2 Method

### 2.1 Concept of BS2-CCNC

The BS2-CCN counter, a modified commercial DMT-CCNC with a newly designed inlet system, measures the activation of

size-resolved CCN with a high-time resolution (Su et al. 2016). Aerosol particles are introduced with a low SAR by wider

inlet and distributed over a continuous range of $S$ in the activation tube whereas aerosol particles of a commercial DMT-CCNC

are forced onto the centerline facing a single $S$. The BS2-CCN system reduces the time required for scanning $S$ and thereby

obtains the κ with a high temporal and size resolution by using a monotonic $F_{act} - S_{aerosol}$ relation. Figure S1 shows the

comparison of supersaturation distribution in the activation tube of CCNC, denoted by $S_{tube}$, between typical CCNC and a

modified CCNC for the BS2-CCN system. $S_{tube}(r)$ is a function of $r$, the radial distance to the centerline of the activation

unit, and is the highest in the centerline (r =0). For a typical CCNC, aerosol particles pass through the centerline in the activation

tube by narrow inlet and laminar sheath flow (Fig. S1a). Particles can be activated as CCN (i.e., $F_{act} = 1$) when $S_{aerosol}$ of

particles is lower than $S_{tube}(r = 0)$ or cannot be activated (i.e., $F_{act} = 0$) when $S_{aerosol}$ is higher than $S_{tube}(r = 0)$. For the

modified CCNC for the BS2-CCN system, aerosol particles are introduced in a broad cross-section of the activation tube

through a wider inlet of a low sheath-to-aerosol flow ratio (Fig. S1b). Therefore, aerosol particles show a monotonic

dependence between $F_{act}$ and $S_{aerosol}$ based on $S_{tube}$ distribution as they are simultaneously exposed to a wide range of S.

The κ value of certain $D_d$ can be derived by the $F_{act} - S_{aerosol}$ relation directly based on κ −Köhler theory. This relation

implies that $S_{aerosol}$ can be directly determined depending on $F_{act}$ and therefore, it is essential to get an accurate calibration

curve (i.e., $F_{act} - S_{aerosol}$ relation).

### 2.2 Experimental setup

Figure 1 shows the schematic plot of the instrumental setup for the BS2-CCN system. The setup consists of an aerosol

classifier, a condensation particle counter (CPC), a modified DMT-CCNC, and other components to maintain and monitor the

working system. The aerosol sample is firstly dried to RH < 30% with an aerosol dryer before entering the aerosol classifier





(TSI classifier 3080). A sensor to monitor the temperature (T), pressure (P), and RH of the aerosol sample is placed in the aerosol flow pathway. A Y-shaped splitter is mounted at the outlet of the aerosol classifier to split the selected monodispersed aerosols into two aerosol flows, CPC (TSI CPC 3772) and modified DMT-CCNC (CCN 100, DMT).

Compared to the commercial DMT-CCNC, a modified DMT-CCNC has a redesigned inlet and flow maintaining system. The newly designed diffusive inlet is used to distribute aerosol samples widely in the activation tube, as detailed in Appendix A.

This new inlet allows for maintaining stable low sheath-to-aerosol flow ratios (SAR), for which monotonic $F_{act} - S_{aerosol}$ relation can then be obtained. Compressed air is used to provide the sheath flow with a HEPA filter in-line to remove all particles in the flow. Its volume flow rate is controlled by a mass flow controller (MFC, Bronkhorst). The aerosol and sheath flows are set to 0.46 L min$^{-1}$ and 0.04 L min$^{-1}$, respectively. The calibration curves with the different flow set in Fig. S2 show that, if the sample flow is set low, the slope between $F_{act}$ and $S_{aerosol}$ gradually decreases due to the narrow $S$ distribution in

the activation tube, making it difficult to obtain a monotonic relationship. It is noted that the total flow in the modified DMT-CCNC is maintained at 0.5 L min$^{-1}$ by vacuum pump with MFC and filter.

For avoiding the water depletion in the activation tube by high particle number concentration, an aerosol humidifier is placed in the aerosol flow pathway to pre-humidify the monodispersed aerosols before it enters the activation tube (dashed line in Fig. 1). Pure water is circulated between the humidifier and a water reservoir by the Bath circulator (Thermo Scientific).

$N_{CCN}$ is recalculated with bin counts data and sample flow as $N_{CCN}$ values recorded in the instrument software are different from what we measure due to the separated flow control system in modified DMT-CCNC. It can be calculated as follows:

$$N_{CCN} = Sum\ of\ Bin\ counts/Sample\ flow\ \times 60 \qquad (1)$$

The 60 is a unit conversion factor since bin counts are given in particle per second while sample flow is in cm$^{-3}$ min$^{-1}$. Particle number concentration (i.e., Number concentration of condensation nuclei, $N_{CN}$) is measured by CPC. $F_{act}$ is calculated as the

ratio of $N_{CN}$ and $N_{CCN}$. Although each of $N_{CN}$ and $N_{CCN}$ data has 1s time resolution and thereby $F_{act}$ data with 1s time resolution are initially available, we use 1-min average data to calculate $F_{act}$ value corresponding to each $D_d$ for the calibration curve.

## 3 Results and discussion

### 3.1 Calibrations

For the calibration of the BS2-CCN system, the goal is to determine the monotonic $F_{act} - S_{aerosol}$ relation, as discussed above. This can be obtained from the size-resolved CCN measurement with pure calibration aerosols, e.g., ammonium sulfate and sodium chloride, in which hygroscopic properties are well known. A specific $S_{aerosol}$ corresponding to each $D_d$ can be calculated from an approximate solution in Eq. (2).

$$S_{aerosol} \approx 100\% \times \left( exp\left( \sqrt{\frac{4A^3}{27\kappa D_d^3}} \right) - 1 \right) \qquad (2)$$



The $F_{act}$ value of each $D_d$ can be measured by the size-resolved CCN measurement, and thereby we can get a calibration curve, $F_{act} - S_{aerosol}$ relation. Figure 2 and 9 show exemplary of calibration curves obtained in this study. We investigate the impact of several factors that may affect the calibration results.

### 3.1.1 $S_{tube}$ distribution

$S_{tube}$ distribution in the activation tube shows the maximum in the centerline (i.e., $S_{max}$) and the minimum at the edge (Fig.
S1). Depending on the temperature gradient, controlled by the temperature difference (dT) between the top and the heated bottom of the activation tube, the distribution of $S_{tube}$ varies, resulting in different shapes of the calibration curve. Figure 2 shows the calibration curves depending on temperature gradients (dT= 6, 8, and 10 K). For calibration experiments, we used an aerosol atomizer to generate particles with diameters of 20 – 300 nm of ammonium sulfate (purity > 99.5%, VWR Chemicals) and sodium chloride (purity > 99.5%, Sigma-Aldrich) solutions. According to Fig. 2, monotonic $F_{act} - S_{aerosol}$
relation is confirmed by the investigated aerosol systems. Good agreements between two calibration aerosols for all three dT conditions show the reproducibility and stability of the BS2-system in measurement CCN activity, supporting its application in the real atmosphere. Large standard deviations at high $F_{act}$ range is mainly caused by the low particle counts $N_{CN}$ at large diameters (e.g., less than 20 cm$^{-3}$).

Although the calibration curve covers the whole range of $F_{act}$, we suggest using only the center part of the curve, from 0.1 to
0.9 of $F_{act}$. This is because steeper slope at low $F_{act}$ and aforementioned low $N_{CN}$ at high $F_{act}$ can introduce large uncertainties in the retrieved $S_{aerosol}$ and κ. Moreover, it is essential to determine an appropriate dT, so that the supersaturation at the centerline, $S_{max}$, is higher than the highest $S_{aerosol}$ of aerosol particles for investigated size range and environment. For the dependence of $S_{aerosol}$, κ and $D_d$, please refer to the κ −Köhler equation (Petters and Kreidenweis, 2007) and exemplary of Fig. 2 in Wang et al. (2015).

**3.1.2 Minimize the effect of doubly/multiply charged aerosols**

Particles of given electrical mobility passed through the DMA are not all singly charged as a DMA extracts particle with a narrow range of electrical mobility rather than a geometric diameter. Multiply (mostly doubly) charged particles with a larger size also penetrate the DMA, accompanying the singly charged particles with a targeted diameter. As the CCN activation of a particle strongly depends on its size, a high fraction of doubly charged particles can directly affect the calibration curve (i.e.,
$F_{act} - S_{aerosol}$ relation).

For solving this problem, Frank et al. (2006) suggested a correction method that subtracts the doubly charged particle distribution in the number size distribution of polydisperse calibration aerosol from $N_{CCN}/N_{CN}$ under the assumption of a bipolar equilibrium charge distribution. Rose et al. (2008) proposed the simple alternative method that calculates the fraction of activated doubly charged particles from the lower level of the plateau in the CCN spectrum, assuming a constant fraction
over the whole particle size range. However, these methods are for the CCN efficiency spectra of typical CCNC using a single





$S$ in the center line of the activation tube. As BS2-CCNC has a wide range of $S$ simultaneously in the activation tube and the number concentration of activated particles is different depending on $S$, it is quite complicated to apply existing methods to experimental data. Here, we compare CCNC responses with and without considering doubly charged aerosol particles and how it depends on the size distribution of calibration aerosols through calculation and experiment results.

We use an activation model that describes the CCNC response to the transferred polydisperse charge-equilibrated particles through an ideal DMA, similar to Petters et al. (2007). Considering electrical mobilities of particles classified by DMA and the fraction of particles carrying $n$ charges (+1, +2) at charge equilibrium, this model calculates an idealized CCN instrument response with an assumed log-normal particle size distribution. When calculating the number of particles that activate as CCN, we need to consider the activation fraction for each aerosol particle size. As the aerosol particles in the BS2-CCN system are

distributed in a broad cross section of activation tube, the activation fraction is calculated by integrating the activation fraction function and flow velocity over the cross-section of the aerosol flow. The detailed calculation procedure is described in Supplement S1.

Figure3 shows the calculated activation fraction ($F_{act\_total} = F_{act\_single} + F_{act\_double}$) for ammonium sulfate aerosolsand the ratio of [+2]/[+1] charges at charge equilibrium of an assumed a log-normal size distribution with $N = 2000\ cm^{-3}, D_g =$

$50\ nm, and\ \sigma_g = 1.4$. Sheath flow ($Q_{sh}$), and aerosol flow ($Q_a$) of DMA were set to 10 lpm and 1.5 lpm, respectively, which are the same as the calibration experiment. It is noted that we used the physico-chemical properties of ammonium sulfate for the calculation and set 0.63% for $S_{max}$ for the $S$ distribution in the activation tube when calculating the activation fraction of aerosol particles. A small plateau exists in an area where the $F_{act}$ is low ($D_p < 40nm$) due to doubly charged particles, however the overall effect of multiply charged particles on $F_{act}$ is not significant and has only a small effect. The maximum

$F_{act\_double}$ in the assumed particle size distribution is about 0.04. According to Fig. 4, the $F_{act\_double}$ varies depending on the particle size distribution. The $F_{act\_double}$ not only increases as the value of $D_g$ increases, but also increases as the $\sigma_g$ increases, even if the $D_g$ is the same. As the $F_{act}$ directly affects the calculation of κ for the BS2-CCN system, the effect of particle size distribution still needs to be considered even though $F_{act\_double} < 0.1$ for $D_g = 60$ nm case. Therefore, when generating calibration aerosols, small $D_g$ and $\sigma_g$ in number size distribution are recommended to minimize the effect of multiply charged

particles on the calibration curve. These effects can also be seen in the calibration experiment using sodium chloride. Figure 5 presents the calibration curves and number size distribution of $N_{CN}$ and $N_{CCN}$ of sodium chloride particles for dT= 10 and 8 K. Sodium chloride that has a high κ value (κ = 1.28) shows variant calibration curves depending on the particle number size distribution, whereas ammonium sulfate shows only small change in the calibration curves (Fig. S3). The $F_{act}$ in calibration curve is higher for larger peak diameter ($D_{peak}$), and accordingly, the gap in the calibration curve between the ammonium

sulfate and sodium chloride increases. Specifically, according to Fig. 5a (dT=10 K), the calibration curve of sodium chloride particle with $D_{peak}$ of 31 nm matches well with that of ammonium sulfate. However, the calibration curve of sodium chloride with $D_{peak}$ of 37 nm and 52 nm are inconsistent with that of ammonium sulfate. This effect is more pronounced at higher supersaturation conditions. Both calculation and experiment results imply that the number size distribution of the generated



particles should be considered, especially when using sodium chloride during calibration, and it is recommended to generate

aerosols with $D_{peak}$ corresponding to an $F_{act}$ less than 0.3. The number size distribution of generated calibration aerosols can be controlled by adjusting the particle concentration.

### 3.1.3 Effect of particle number concentration

As the modified CCNC enlarges the cross-section of aerosol flow, more particles can enter the column compared to a in the standard DMT-CCNC. The consumption of water vapor in the column by a large number of particles can change the

distribution of the supersaturation and thereby influence the measured number fraction of activated particles. Therefore, as a simple test, the $F_{act}$ values of three different sizes of ammonium sulfate particles (60, 80, and 120 nm) are measured under the different particle number concentrations. In this experiment, dT is set to be 7.7 (S = 0.6%). According to Fig. 6, we can see that $F_{act}$ decreases with a rate of about 0.02/(100 cm$^{-3}$) with the increase of the particle number concentration. Specifically, $F_{act}$ decreases by about 2.6% (60 nm), 1.6% (80 nm) and 1.1% (120 nm) per increase of 100 cm$^{-3}$ in the number of particles.

It is noted that the decrease with particle number is calculated for a particle number concentration of 300 cm$^{-3}$ of ammonium sulfate, and the decrease is expected to be greater if the number concentration increases. The decreasing rate increases slightly as the particle size decreases. These results imply that the number concentration of calibration aerosols can affect the calibration result, and we can get consistent results even in very low number concentrations from this BS2-CCNC setup.

For examining the necessity of a humidifier to avoid water consumption in the activation tube, an aerosol humidifier is installed

additionally in the aerosol flow pathway. The setup is described in Fig. 1, and the part is marked by the blue dashed line. Monodispersed aerosols are pre-humidified by the humidifier that is composed of a Nafion tube and a bath circulator before the aerosol flow enters the BS2-CCN system. Particle number concentrations are controlled within the range of ~ 300 cm$^{-3}$. Figure 7 shows the calibration curves for three dT conditions (dT=6,8, and 10 K) with a humidifier (blue dots) and without a humidifier (black dots). For dT=6 K, the calibration curves with the humidifier system (WH) and without the humidifier system

(NH) are almost identical. Although $F_{act}$ values of WH are slightly higher than those of NH under higher dT conditions (i.e., dT= 8 and 10 K), differences are not significant as number concentrations of calibration aerosol are not so high. In other words, a compact instrumental setup without the pre-humidifier system is sufficient for the BS2-CCNC calibration experiment as well as the measurement if aerosol particles are kept below ~ $3\times10^2$ cm$^{-3}$. Otherwise, we need a pre-humidifier system for high aerosol number concentration condition to avoid the decrease of $F_{act}$. It is noted that particle concentrations below ~$3\times10^3$

cm$^{-3}$ are recommended to avoid counting error for calibration experiment of typical DMT-CCNC (Rose et al., 2008). We usually observe the low number concentration for aerosol particles during the size-resolved CCN measurement and $F_{act}$ is not highly variable within that range. However, we still need to consider this effect, especially for the region of high number concentration and/or high-number concentration cases like new particle formation (NPF) events and the transport of pollution.



## 3.2 Fitting procedure of calibration curve

As selected diameters for calibration are limited and cannot cover the whole $F_{act}$ values, the curve fitting procedure is necessary. The equation for curve fitting (Eq.3), the relationship between $F_{act}$ and $S_{aerosol}$, can be derived based on Eq. (S8) and cosine function of $S_{tube}$ distribution in Supplement S1.

$$F(x) = a \times acos(b \times x) - c \tag{3}$$

where $F(x)$ and $x$ correspond to $S_{aerosol}$ and $F_{act}$, respectively. Coefficients of $a$, $b$, and $c$ are what we need to be obtained from the curve-fitting procedure. They are calculated using a non-linear least square method (MATLAB curve fitting toolbox 3.5.8). It is noted that the data less than 0.05 of $F_{act}$, showing the large discrepancy between reference curves and experimental data, are excluded for curve fitting.

Figure 8 shows fitting curves and experimental data from ammonium sulfate particles for three dT conditions (dT = 6, 8, and 10 K). Coefficients and goodness of fit for each curve are presented in Table 1. For assessing the goodness of fit, three statistical parameters were used: Error sum of squares (SSE), coefficient of determination ($R^2$), and root mean square error (RMSE). According to Fig. 8 and Table 1, fitting curves cover most of the experiment data, and good fitting results are shown for all three dT conditions. It implies that Eq. (3) covers the experimental data well and appears suitable for the calibration curve of the BS2-CCN system. Also, $S_{aerosol}$ corresponding to the $F_{act}$ can be directly obtained through this curve. Furthermore, we can calculate the effective particle hygroscopicity parameter, κ, based on κ − Köhler theory (Petters and Kreidenweis, 2007). Figure S4 presents κ values, which correspond to the $F_{act}$ values of a particle ranged from 50 nm to 150 nm. They are calculated with a fitting curve equation of dT=8 K condition shown in Fig. 8 and Table 1.

As $F_{act}$ values obtained from the size-resolved CCN measurement use 1-min average data, the calibration curve could have the range, not a single curve line. To consider the range of $F_{act}$, we add two more calibration curve lines applying data points of $F_{act\_low}$ (average - standard deviation, μ-σ) and $F_{act\_high}$ (μ+σ). Figure 9 shows the range of the calibration curve for the dT=8 K condition. All of the calibration curves are obtained using Eq. (3) for curve fitting. When the $F_{act}$ is measured at a certain $D_p$, three κ values, denoted as $κ_{high}$, $κ_{avg}$ and $κ_{low}$, can be calculated based on these three curves. Specifically, $κ_{high}$ is derived from the curve that is obtained from data points of $F_{act\_low}$, and vice versa.

## 4. Evaluation and application of BS2-CCN system

To evaluate the BS2-CCN system, we performed two inter-comparison experiments between BS2-CCN and standard DMA-CCN measurement (denoted as 'DMA-CCN') with a laboratory-generated aerosol mixture and ambient aerosols. The DMA-CCN measurement is widely used for the size-resolved CCN measurement in the aerosol community to examine the CCN activity (Cai et al., 2018; Deng et al., 2011; Moore et al., 2011; Rose et al., 2011; Pöhlker et al., 2016, 2018; Thalman et al., 2017 and references therein). For DMA-CCN measurement, critical diameter ($D_c$) that $F_{act}$ becomes 0.5 is determined by scanning $D_p$ under a given constant supersaturation (i.e., $D_p$ scan) and κ can be calculated with a given S and $D_c$ based on κ −





Köhler theory (Petters and Kreidenweis, 2007). In other words, in DMA-CCN measurement, we can obtain single $D_c$ value for each $D_p$ scan and convert it to single κ, but in BS2-CCN measurement, we can obtain κ values at all selected particle sizes of each $D_p$ scan by converting $F_{act}$ values using the calibration curve. Figure S5 presents the schematic plot of the instrumental setup for inter-comparison experiments. Specifically, selected monodisperse aerosol flows by the DMA (TSI classifier 3080) are split into three parallel lines and fed into CPC, modified CCNC (CCN-100, DMT for BS2-CCNC), and DMT-CCNC (CCN-200, DMT for DMA-CCNC). Before entering the DMA, all aerosol particles are dried (RH < 30%) by the Nafion tube aerosol dryer and neutralized by Krypton-85 (Kr 85) bipolar charger. All instruments were installed in the laboratory at the Max Planck Institute for Chemistry, Mainz, Germany. A detailed set of experiments for each inter-comparison experiment is described in the following section.

**4.1 Ammonium sulfate and succinic acid mixture**

We used an atomizer to generate nanoparticles with diameters of 30 – 160 nm by spraying a mixed solution of succinic acid and ammonium sulfate. This mixture was chosen to mimick a typical atmospheric aerosol composed of ammonium sulfate and organic acids. For the mixture, molar ratios of ammonium sulfate and succinic acid used in this study were 1:1, 3:1, and 1:3. The experiments were performed under dT=8 K (0.63 % S) condition. The sample flow rates of CCNCs were 0.04 lpm and 0.46 lpm for the DMA-CCN and BS2-CCN measurements, respectively. The total flow rate (sample + sheath) was 0.5 lpm for both instruments. As the κ of ammonium sulfate is higher than that of succinic acid (i.e., more hygroscopic), a mixture of ammonium sulfate and succinic acid in a ratio of 3:1 shows the highest κ value among three mixtures and vice versa. Table 2 presents details of the comparison for all three solutions. The κ values of the DMA-CCN measurement for all three solutions are within the range of κ for the BS2-CCN measurement, but is slightly lower (less than about 10 % of relative deviation) than the $\kappa_{avg}$. These results infer that we can measure κ of aerosol mixture quantitatively well from the BS2-CCN measurement compared to the existing measurement method, DMA-CCN measurement.

**4.2 Ambient aerosol measurement**

An inter-comparison experiment with ambient aerosols was performed from 13 July to 16 July 2020. The instrumental setup and flow system were the same as in Section 3.1. For $D_p$ scan (fixed $S$), 19 dry diameters of 40 – 250 nm were selected for each scan. Each scan took a total of 22 minutes, including 1 min for each diameter and 3 min for stabilization. The $S$ of CCNC was set to be 0.63 % (dT=8 K) for BS2-CCN measurement and 0.4% for DMA-CCN measurement. The $S$ of DMA-CCN measurement was set slightly lower because the comparison with κ of BS2-CCN measurement is difficult if the critical diameter is too small, which is obtained from the DMA-CCN measurement. It is noted that κ values from BS2-CCN measurement were calculated based on the fitting curve for dT=8 K, as shown in Fig. 9 and Table 1. Figure 10 shows the times series of $F_{act}$ and κ distribution of BS2-CCN measurement. Here $\kappa_{avg}$, derived from $F_{act\_avg}$, is referred to as κ. For κ calculation, high $F_{act}$ (> 0.85) and low $F_{act}$ (< 0.1) data are excluded. $F_{act}$ distribution showed a clear size-dependency, low



$F_{act}$ for small particles and high $F_{act}$ for large particles. Since the κ value is calculated based on the $F_{act}$ value, the κ value appeared to be increasing at the time when the $F_{act}$ was increased, and vice versa. The average κ values of particles for the BS2-CCN measurement exhibit diurnal variability that increases during the daytime and decreases at nighttime within the range of 0.11 to 0.32 (Fig. 10 and Fig. S6) and has an average value of 0.18. For DMA-CCN measurement, average $D_c$ and κ was about 80 nm and 0.17, respectively, during the measurement under the 0.4 % $S$ condition. Figure S7 presents the average CCN efficiency spectra and the cumulative particle hygroscopicity distribution, $H(\kappa, D_d)$, of DMA-CCN measurement. Particularly, the κ values of a significant portion of particles were distributed between 0.1 and 0.3, which was consistent with the result of the BS2-CCN measurement. Figure 11 presents the hourly-averaged κ value of BS2-CCN and DMA-CCN measurement. Unlike BS2-CCN measurement results, which show κ distribution of various particle sizes, DMA-CCN measurement allows a single κ value to be produced per each $D_p$ scan cycle. Therefore, as shown in Fig. 11, we used κ value of BS2-CCN measurement by selecting the particle diameter close to the average $D_c$ of the DMA-CCN measurement for the inter-comparison. Compared to the κ values of the DMA-CCN measurement, those from the BS2-CCN measurement showed a good agreement, keeping up with the increasing and decreasing variability. Additionally, a direct inter-comparison was carried out through the 1:1 scatterplot between the κ of BS2-CCN and DMA-CCN measurements in Fig. 12. All the detailed inter-comparison results, including the ratio between κ of DMA-CCN and BS2-CCN measurement and goodness of fit of the linear regression line for each scatter plot, are presented in Table 3. It is noted that $\kappa_{high}$, showing excessively high value compared to that of DMA-CCN measurement, are excluded in Fig. 11 and 12. We can conclude through the good agreements of κ value as well as results of three different statistical values to judge the goodness of fitting, including a residual sum of squares, Pearson's r, and $R^2$ that we can obtain reliable and quantitative κ data, as well as the κ variability from BS2-CCN measurement compared to the existing DMA-CCN measurement. The possible reason for the discrepancy of κ between BS2-CCN and DMA-CCN measurement is the multimode κ distribution. According to Fig. S7, we can infer that aerosols were externally mixed, not a single mode. The BS2-CCN system alone cannot resolve bimodal or multimodal κ distribution, and thereby when particles at a certain size are externally mixed, lower $F_{act}$, resulting in lower κ, can be observed. In particular, κ values of BS2-CCN measurement are slightly higher than those of DMA-CCN measurement between 15 and 16 July when externally mixed aerosols were frequently observed. Su et al. (2016) pointed out through the simulation that BS2-CCN measurement can underestimate κ when particles are externally mixed. The uncertainty of $F_{act}$ values for each measurement caused by the uncertainty of each CCNC can be another possible reason for the discrepancy. The $F_{act}$ value is essential for both measurements as κ is directly determined by $F_{act}$ for BS2-CCN and $D_c$ of a DMA-CCN measurement can be changed depending on $F_{act}$. The selected diameters of BS2-CCN measurement for comparison in Fig. 11 and Fig. 12 do not perfectly match with the $D_c$ of DMA-CCN measurement because $D_c$ in this study is not fixed due to the '$D_p$ scan' method, scanning $D_p$ with a constant $S$. The difference of time resolution can also be a cause of the discrepancy, although the impact may not be significant in this study. As we use 1 min average data per particle size, we can obtain κ values for every 1 minute for the BS2-CCN measurement but only every 19 minutes for the DMA-CCN measurement. Therefore, a difference κ is expected if there



are fast changes of the aerosol. Nevertheless, good correlation and correspondence of κ values between DMA-CCN and BS2-CCN measurement from both inter-comparison experiments infer that we can obtain a high time resolution with reliable hygroscopicity data from the BS2-CCN system. These powerful advantages allow for applying the BS2-CCN system preferably to ship and aircraft measurements requiring high time-resolution, as well as ground-based measurements for κ distribution with a broad particle size range.

## 5 Summary and conclusion

In this study, we implement a new concept for the design of CCN counters, a broadening supersaturation scanning BS2-CCN system, for rapid hygroscopicity measurement and describe how to calibrate this system. Compared to the typical CCN counters, particles are exposed to a range of $S$ simultaneously in an activation tube with a newly designed inlet and low sheath-to-aerosol flow ratio (SAR). Through this system, we can obtain a monotonic relation between $F_{act}$ and $S_{aerosol}$. Based on the

$\kappa - K\ddot{o}hler$ theory, κ can be derived directly through the calibration curve (i.e., $F_{act} - S_{aerosol}$ relation) when we measured the $F_{act}$ value at a certain $S$.

For calibration, ammonium sulfate and sodium chloride, representative calibration aerosols for CCNC, are used under three different dT conditions. It can be inferred from consistent results between two calibration aerosols as well as reference curves for all three dT conditions that the experimental setting of the BS2-CCN system, suggested in this study, is appropriate and

can apply to the real measurement. We also examine factors that can affect the calibration curves. In the first, $S_{max}$, the maximum $S$ in the activation tube, determine $S_{tube}$ distribution in the activation tube and the shape of the calibration curve changes accordingly. The range of particle size and κ value that we can measure depends on the $S_{max}$ value. Specifically, a high $S_{max}$ can cover the wide range of particle size and κ, but the steeper slope of the calibration curve can lead to high sensitivity of $S_{aerosol}$ corresponding to the measured $F_{act}$. Therefore, depending on the particle size and environment to be

investigated, we need to determine an appropriate $S_{max}$. Calculation and experimental results confirm that multiply charged particles have a small but measurable effect on the $F_{act}$ value and show that the effect depends on the size distribution of the particles. For minimizing the effect of multiply charged particles, small $D_g$ and $\sigma_g$ in number size distribution are recommended when generating calibration particles. This effect is more pronounced for sodium chloride with very high hygroscopicity and/or higher dT conditions. Lastly, we examine the effect of particle number concentration on the calibration

curve. The activated particle number fraction decreases with a rate of about 0.02/(100 cm$^{-3}$) within ~ 300 cm$^{-3}$ of ammonium sulfate particles, and the decreasing rate is expected to be much higher when the concentration is higher due to the water consumption in the activation tube. It implies that we need to generate fewer particles for the calibration experiment of the BS2-CCN system compared to that of typical CCNC. Particles below ~ $3 \times 10^2$ cm$^{-3}$ are recommended for generating calibration aerosols. If the number of particles is high, the pre-humidifier system is helpful to avoid decreasing $F_{act}$.

As selected particles for the calibration experiment cannot cover the whole $F_{act}$ range, the curve fitting procedure is essential. We propose the equation for curve fitting (Eq.3) based on the equation of $S_{tube}$ distribution. It is noted that the data lower than





0.05 of $F_{act}$ are excluded. Good results of statistical parameters to judge the goodness of fit is shown for all three different dT conditions. With these curves, we performed two inter-comparison experiments between DMA-CCN and BS2-CCN measurement for evaluation; Laboratory generated aerosol mixture and ambient aerosol measurement. Firstly, κ values of the

mixture of ammonium sulfate and succinic acid with three different molar ratios were compared. The κ values of the BS2-CCN measurements agree well with those of DMA-CCN measurement. For ambient aerosol measurement, $S$ of CCNC was set to be 0.63 % (dT=8 K) for BS2-CCN measurement and 0.4% for DMA-CCN measurement. For BS2-CCN measurement, κ distribution between 70 nm and 120 nm in diameter showed a size-dependency, low κ at small particle and high κ at the large particle, and a distinct diurnal variability that increases during the daytime and decreases during the nighttime. Also, the κ

values of BS2-CCN measurement corresponded and correlated well with those from DMA-CCN measurement. It can be concluded from these results that the BS2-CCN system can measure κ quantitatively well compared to the existing measurement method and even can measure a broad range of κ distribution with high-time resolution.

The BS2-CCN system, a simple modification of the commercial design of CCNC, uses constant supersaturation and flow during the measurement, so it is technically simple, and thereby we can obtain stable data with a high-time resolution. And

low SAR of the BS2-CCN system provides sufficient counting statistics for size-resolved measurement, in which particle concentrations are generally low. Besides, the calibration of the BS2-CCN system is not complicated and has many similarities compared to the existing method, so it is easy to apply. The advantages of the fast response and stability, as well as the relatively simple calibration method of the BS2-CCN system, make it possible to apply not only to long-term observation but also extensive measurements including aircraft, ship, and ground. Lastly, as the hygroscopicity can be used as a proxy for the

chemical composition, our method can also serve as a complementary approach for fast and size-resolved estimation of aerosol chemical composition.





**Data availability**

Data can be accessed by contacting the corresponding author.

**Author contributions**

HS and YC had the initial idea, and HS, YC, and NM designed the BS2 instrument. TFM provided the CCNC instrument and discussed the new system. NK organized and performed all experiments. MP provided the instrument for the inter-comparison experiment. OK supported inter-comparison experiment. TK provided technical support for experiments. NK wrote the paper. All coauthors discussed and results and commented on the paper.

**Competing interests**

The authors declare that they have no conflict of interest.

**Acknowledgments**

This work was supported by the Max Planck Society (MPG).





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





## Appendix A: Newly designed diffusive inlet

The newly designed diffusive inlet is comprised of the main body (**101**) and a sheath flow straightener (**102**), as shown in Fig. A1. The main body of the diffusive inlet is constructed from the conductive material. An aerosol inlet (**103**) is placed at the center top of the main body. Downstream of the aerosol inlet has a funnel-shaped region (**105**) where the cross-section of the aerosol is smoothly expanded. The angle (**106**) of the wall of the funnel-shaped region is small enough to keep a laminar flow. The wall of the funnel-shaped region is polished to minimize air turbulence and particle deposition. The inlet of sheath air

(**104**) is placed at the side of the main body. A flow straighter (**102**), located at the down-stream of the sheath air inlet, is made up of a single or double screen of fine nylon mesh to straighten the sheath flow and lead to a laminar flow. This inlet is mounted at the top of the column of the activation tube. At the lower end of the main body, there are two rubber O-rings (**107**) to keep the activation tube air-tight.



**Table 1. Coefficients and goodness of fit for calibration curves for three different dT conditions (dT= 10, 8, and 6 K). Three different statistical values, including Error sum of squares (SSE), coefficient of determination ($R^2$), and root mean square error (RMSE), are used to judge the goodness of fit.**

| Equation | $F(x) = a \times acos(b \times x) - c$ | | |
|---|---|---|---|
| **dT** | **dT=10** | **dT=8** | **dT=6** |
| *Coefficients (with 95% cofidence bounds) | a =0.5097 (0.501, 0.5183) <br> b =0.9912 (0.987, 0.9954) <br> c =9.133e-13 (fixed at bound) | a =0.3895 (0.3601, 0.419) <br> b =0.9892 (0.9635, 1.015) <br> c =-0.005699 (-0.04233, 0.03093) | a =0.3221 (0.3013, 0.343) <br> b =0.9722 (0.9429, 1.002) <br> c =0.05177 (0.02362, 0.07991) |
| *Goodness of fit | SSE = 0.002435 <br> $R^2$ = 0.9961 <br> RMSE = 0.01274 | SSE = 0.00135 <br> $R^2$ = 0.9967 <br> RMSE = 0.009187 | SSE = 0.0002855 <br> $R^2$ = 0.9987 <br> RMSE = 0.004363 |

*Coefficients and goodness of fit were calculated by MATLAB curve fitting toolbox 3.5.8.



**Table 2.** $\kappa_{avg}$ **from DMA-CCN and** $\kappa_{low}$, $\kappa_{avg}$, **and** $\kappa_{high}$ **from BS2-CCN measurement for dT=8 K (S = 0.63%) condition for mixtures of ammonium sulfate and succinic acid.** $\kappa_{low}$, **and** $\kappa_{high}$ **are derived from the range of calibration curves in Fig. 9.**

|  | DMA-CCN | BS2-CCN | | |
|---|---|---|---|---|
|  | $\kappa_{avg}$ | $\kappa_{low}$ | $\kappa_{avg}$ | $\kappa_{high}$ |
| **AS:Su = 1:1** | 0.176 | 0.168 | 0.186 | 0.200 |
| **AS:Su = 3:1** | 0.255 | 0.253 | 0.285 | 0.311 |
| **AS:Su = 1:3** | 0.121 | 0.118 | 0.132 | 0.143 |





**Table 3. The average and standard deviation of ratio between κ of DMA-CCN and BS2-CCN measurement and goodness of fit for linear regression line. Three different statistical values, residual sum of squares, Pearson correlation coefficient and coefficient of determination ($R^2$), are used to evaluate the goodness of fit.**

|  | $F_{act\_low}$ | $F_{act\_mean}$ | $F_{act\_high}$ |
|---|:---:|:---:|:---:|
|  | (μ-σ) | (μ) | (μ+σ) |
| **Ratio** ($\kappa_{BS2-CCN}$ / $\kappa_{DMA-CCN}$) | 1.17±0.17 | 1.05±0.12 | 0.98±0.10 |
| **Residual Sum of Squares** | 0.04 | 0.02 | 0.02 |
| **Pearsons's r** | 0.75 | 0.83 | 0.86 |
| **$R^2$** | 0.57 | 0.70 | 0.74 |



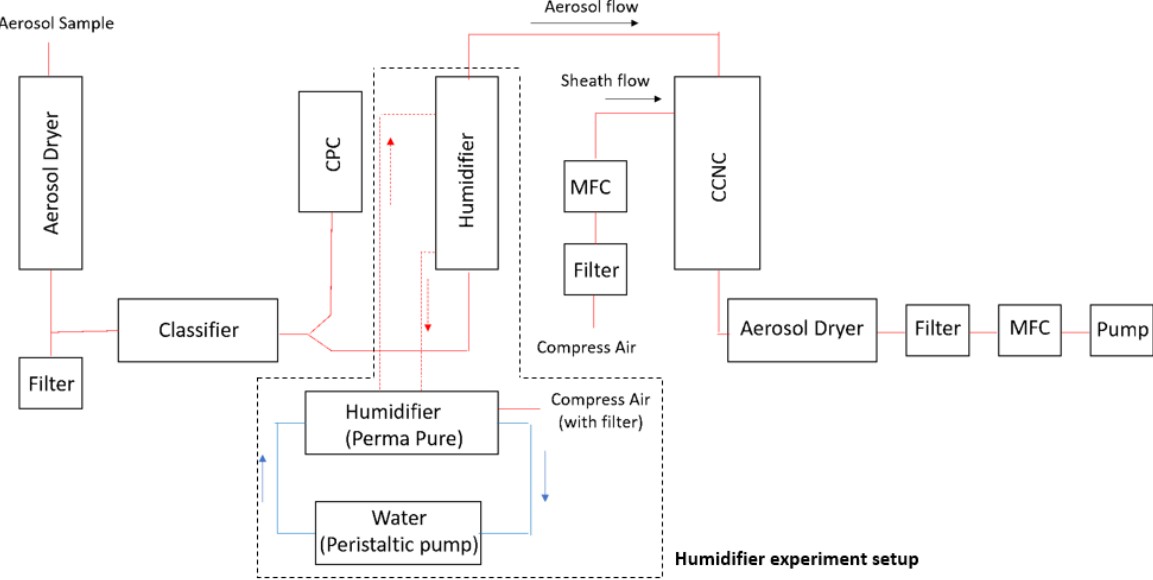

**Figure 1: Schematic plot of a broad scanning supersaturation cloud condensation nuclei counter (BS2-CCN) system. An additional setup marked with dashed line is for a humidifier experiment.**



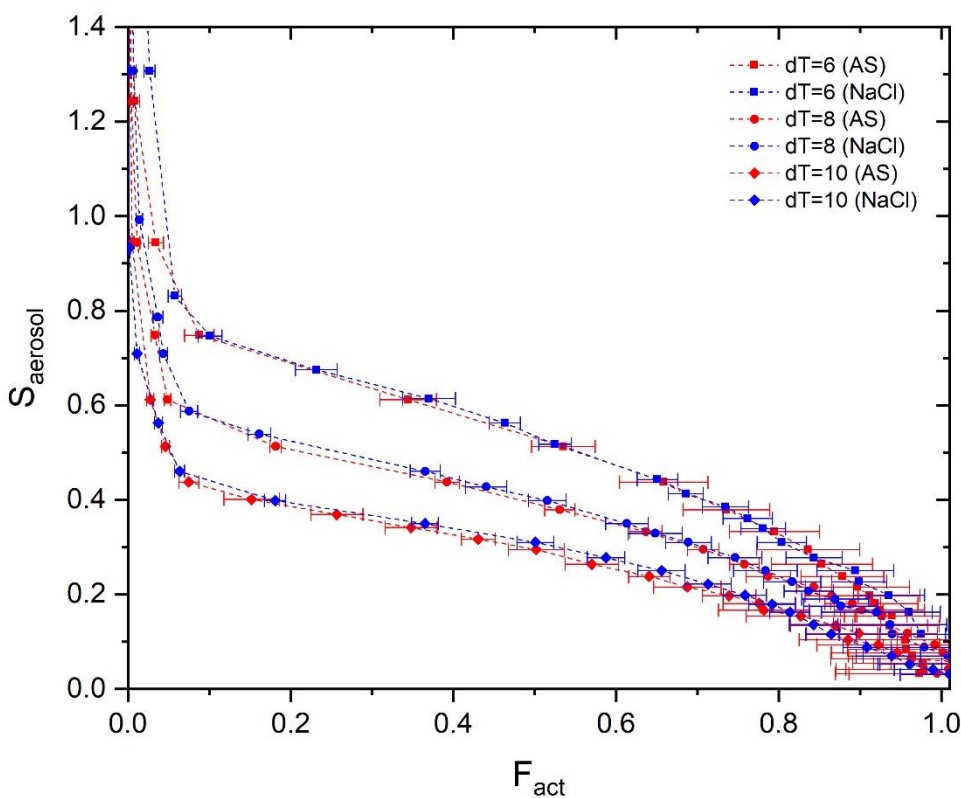

**Figure 2: Calibration curves ($F_{act} - S_{aerosol}$) for three different T gradients (dT = 6, 8, and 10 K) with ammonium sulfate (red) and** 510 **sodium chloride (blue) particles.**

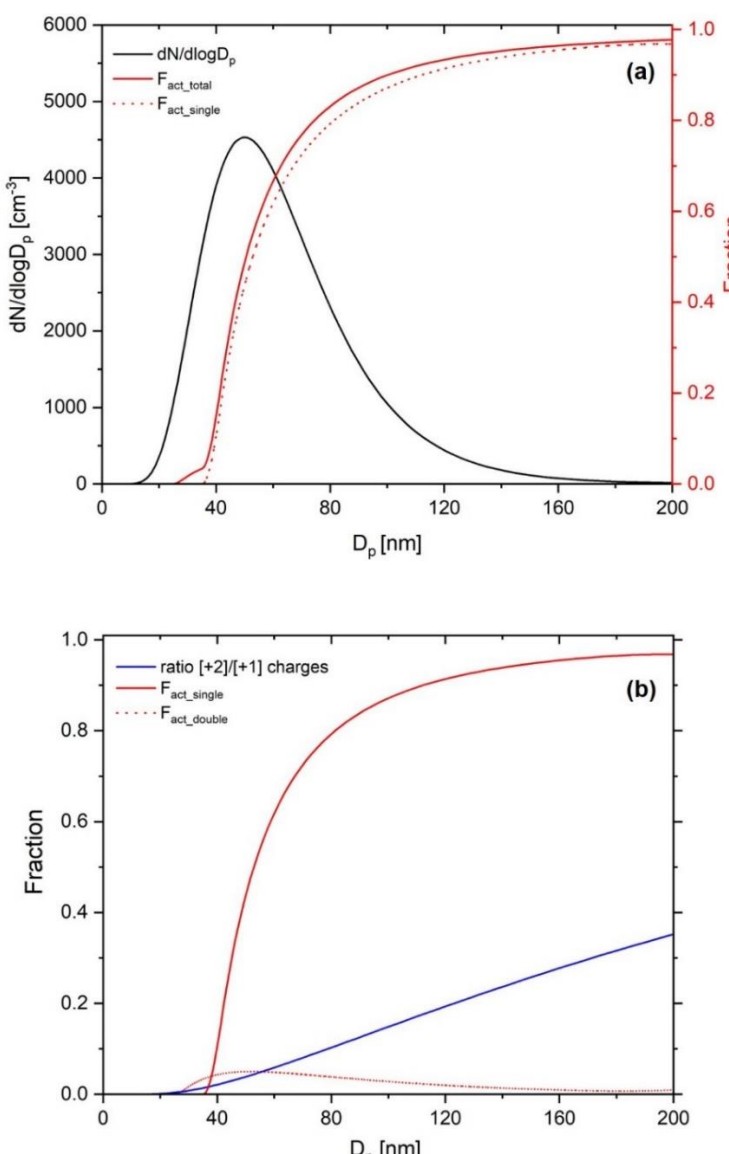

**Figure 3: Calculated ideal activation fraction for log-normally distributed, charge-equilibrated particles transmitted BS2-CCNC system. Shown are (a) assumed log-normal particle size distribution (black solid line, left ordinate, $N = 2000\ cm^{-3}, D_g = 50\ nm, and\ \sigma_g = 1.4$), total activation fraction (red solid line), activation fractions by singly charged particles (red dashed line) and (b) activation fraction by singly charged particle (red solid line) and doubly charged particles (red dashed line), and the ratio of [+2]/[+1] charges (blue solid line), which refers to $f(D, n = +2)/f(D, n = +1)$ with mobility diameter at charge equilibrium. $f(D, n)$ is the fraction of particle carrying $n$ charges at charge equilibrium by Wiedensohler (1988).**





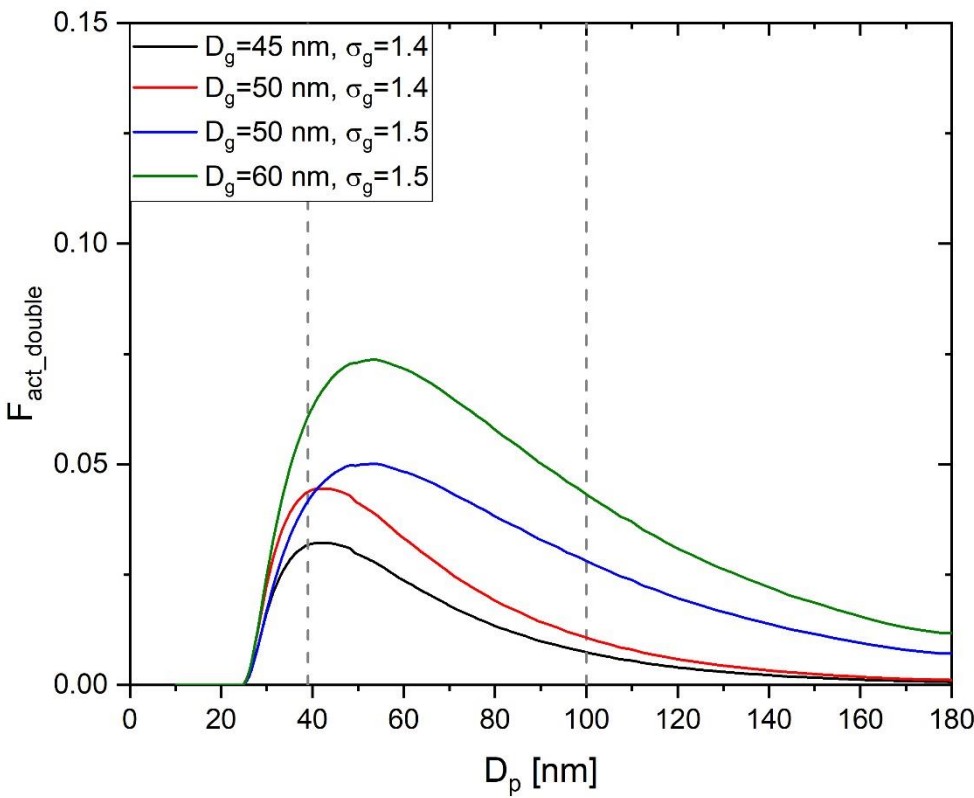

**Figure 4: Activation fractions by doubly charged particles ($F_{act\_double}$) for variant particle size distributions. Information of each particle size distribution is presented in the legend of the figure.**





**Figure 5: Calibration curves of ammonium sulfate and sodium chloride for (a) dT=10 K and (c) dT=8 K and number size distribution**
**of $N_{CN}$ and $N_{CCN}$ for sodium chloride particle for (b) dT=10 K and (d) dT=8 K.**

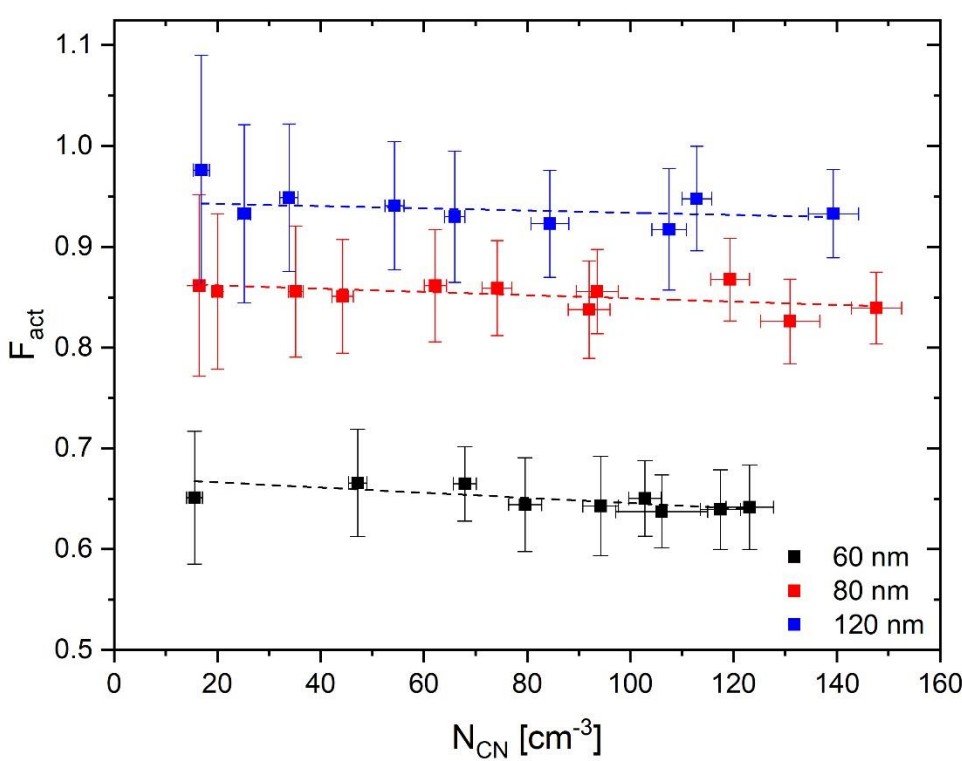

**Figure 6: Average and standard deviation (error bar) of $F_{act}$ depending on the number concentration $N_{CN}$ for 60 nm (black), 80 nm (red) and 120 nm (blue) of ammonium sulfate particle under the dT=7.7 K (0.6% S) condition. Dashed lines indicate linear regression lines.**



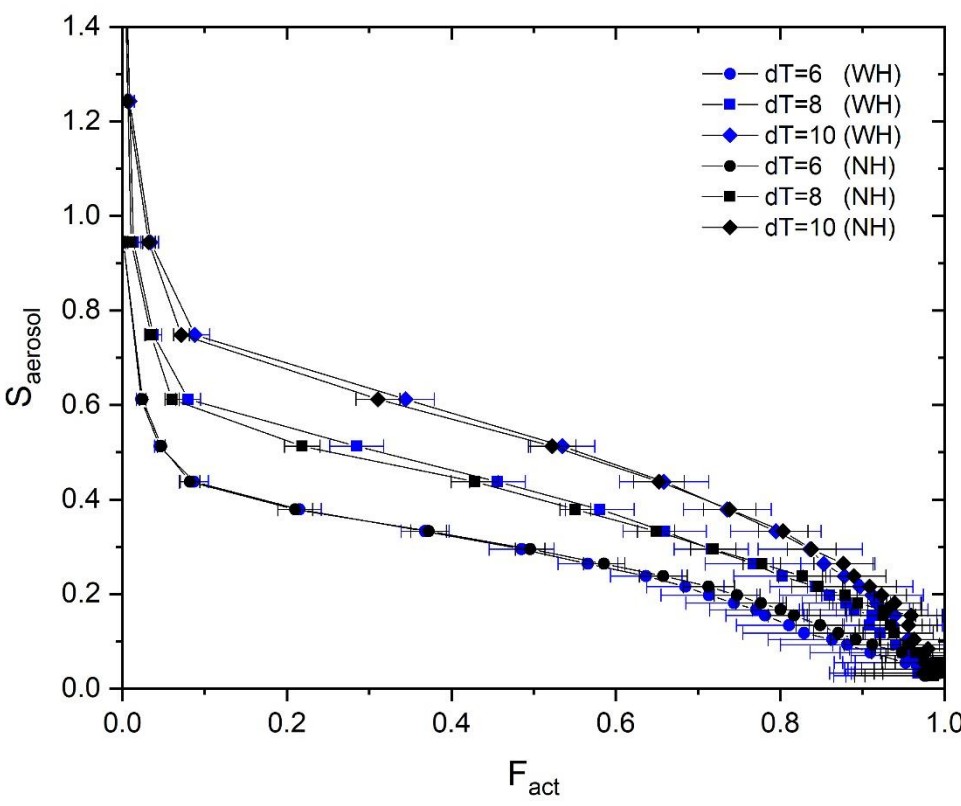

**Figure 7: Calibration curves ($F_{act} - S_{aerosol}$) for three different T gradients (dT = 6 (circle), 8 (square), and 10 (diamond) K) with humidifier system (WH) and without humidifier system (NH).**




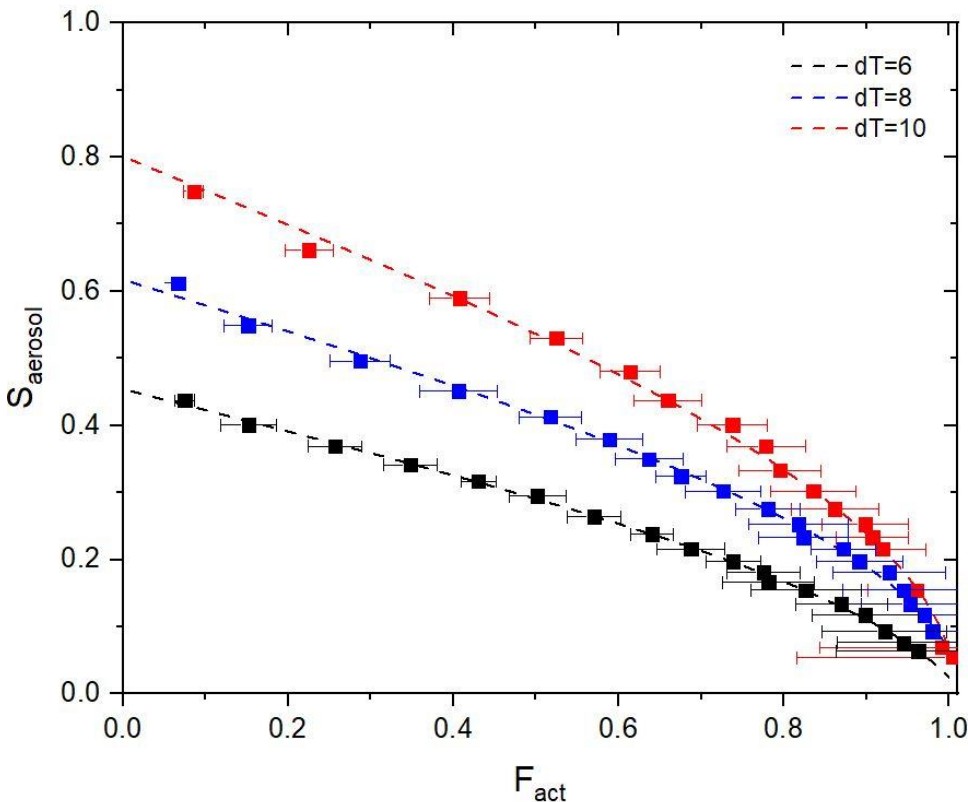

**Figure 8: Experimental data (square dot with error bar) and fitting curves (dashed lines) for three different dT conditions (dT = 6 (black), 8 (blue), and 10 K (red)). Ammonium sulfate particles are used for calibration.**



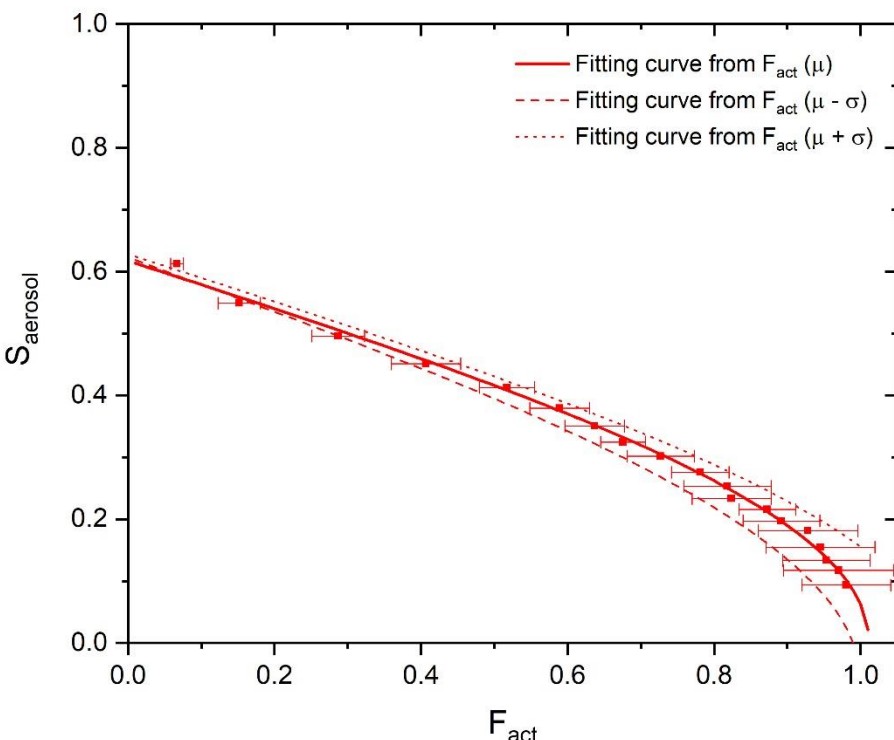


**Figure 9: Experimental data point (square dot with error bar) and fitting curves for dT=8 K (S=0.63%) condition. Solid line is obtained from data of $F_{act}$ (μ). Dotted and dashed fitted line are obtained from data of $F_{act\_high}$ (μ + σ) and $F_{act\_low}$ (μ − σ). μ and σ indicate average and standard deviation of $F_{act}$, respectively.**

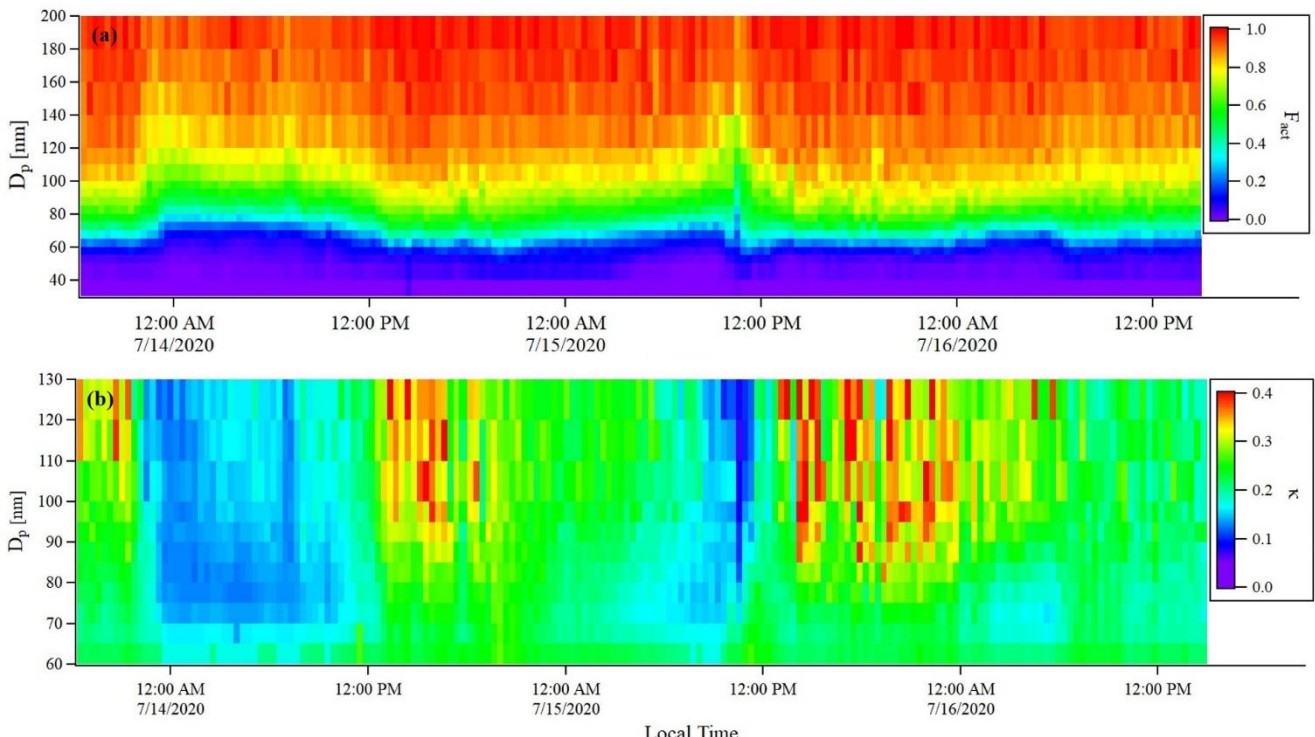


**Figure 10: Time series of (a) $F_{act}$ and (b) κ distribution of BS2-CCN measurement for 0.63% S (dT=8 K) condition. Measurement period was 13 July – 16 July 2020.**





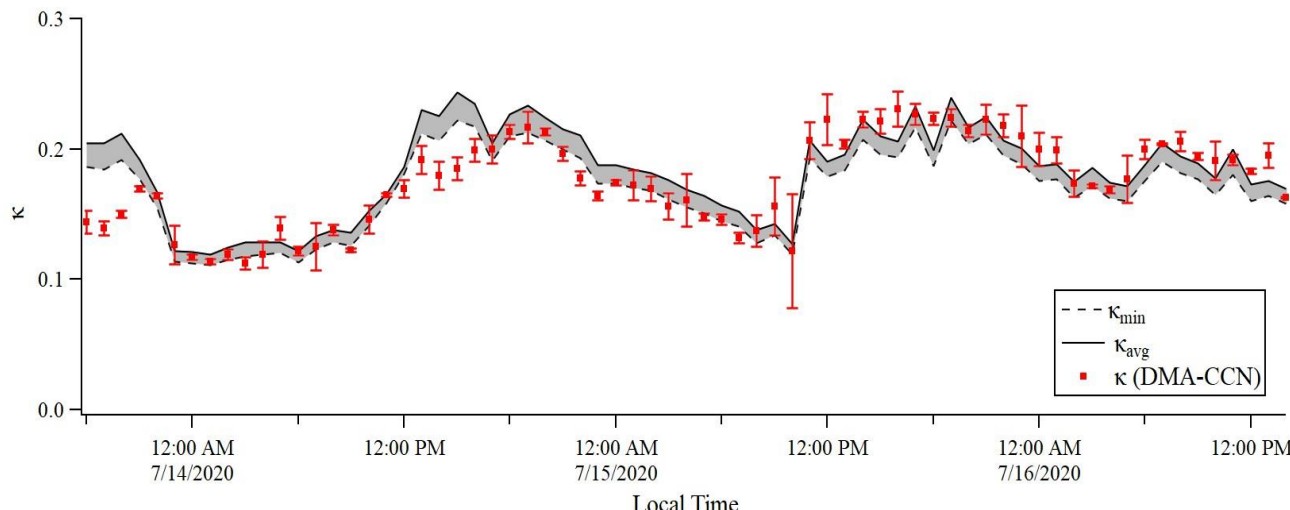

**Figure 11: Time series of hourly averaged κ values for DMA-CCN (red dots, bar indicates the standard deviation of κ) and BS2-CCN (grey shaded area for the range between $\kappa_{avg}$ (black solid line) and $\kappa_{low}$ (black dashed line)) measurements. The $\kappa_{high}$ is excluded in this figure.**

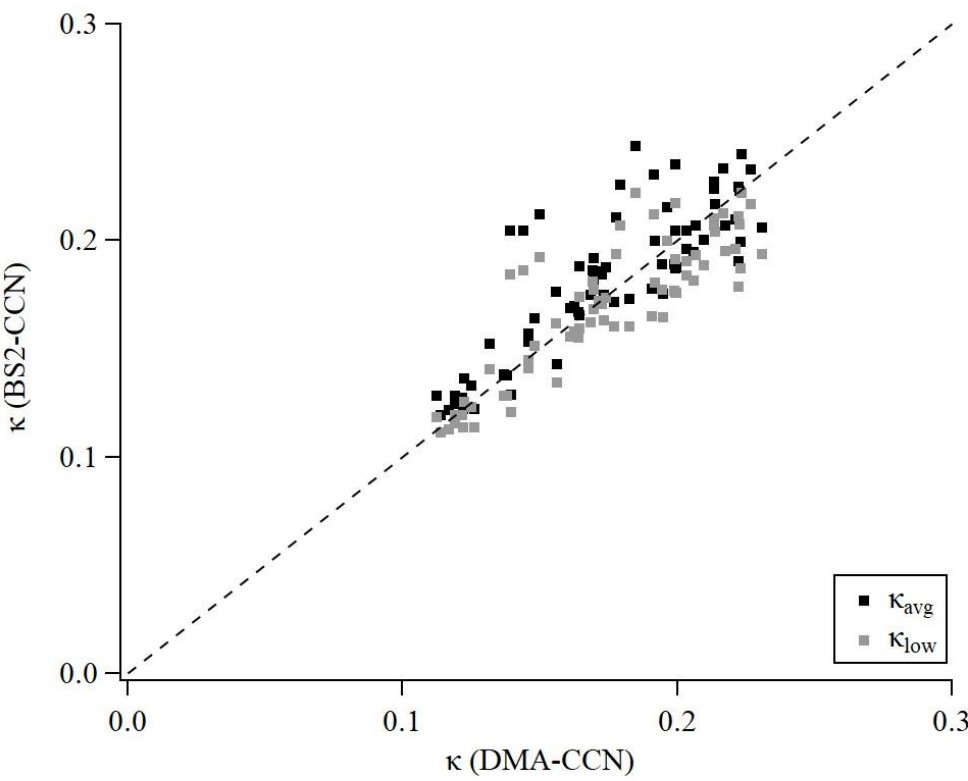

**Figure 12: Scatterplot of κ between DMA-CCN and BS-CCN measurement. Black square and grey square dots indicate $\kappa_{avg}$ (calculated from the calibration curve of $F_{act}$ (μ)) and $\kappa_{low}$ (calculated from the calibration curve of $F_{act\_high}$ (μ + σ)) for the BS2-CCN measurement, respectively. The κ value of BS2-CCN measurement was selected for that in the diameter adjacent to the critical diameter of DMA-CCN measurement for the comparison. The black dashed line is a 1:1 line for clarity.**






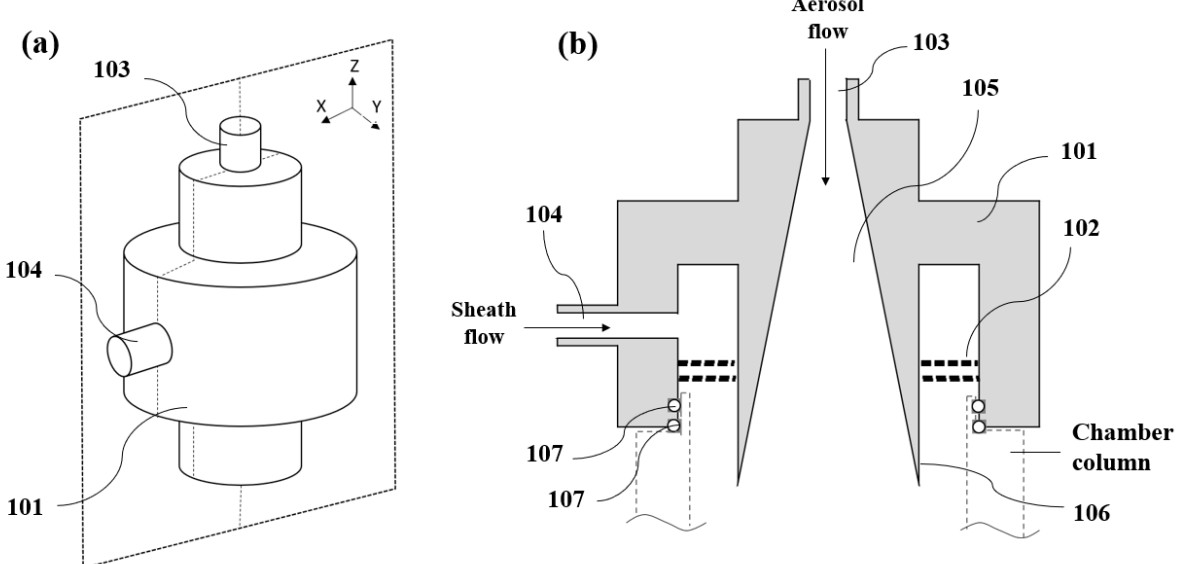

**Figure A1: (a) Front perspective view of an embodiment of the diffusive inlet. (b) Longitudinal sectional (the cross section in X-Z surface) view of Fig.A1 (a). Each of numbers in the figure is as follows: main body (101), a sheath flow straightener (102), an aerosol inlet (103), a funnel-shaped region (105) where the cross-section of the aerosol is smoothly expanded, the angle (106) of the wall of the funnel-shaped region, the inlet of sheath air (104) at the side of the main body and two rubber O-rings (107) at the lower end of the main body to keep the activation tube air-tight.**