# Peer review of "Calibration and evaluation of broad supersaturation scanning (BS2) cloud condensation nuclei counter for rapid measurement of particle hygroscopicity and CCN activity"

_Atmospheric Measurement Techniques, 2021_

## Referee Comment (RC1)

This manuscript presents a novel broad supersaturation scanning CCN (SB2-CCN) system, which can measure the CCN activity with a high time resolution. Overall, this manuscript clearly explained the set-up and calibration of the instrument, and applied it in the field measurement. This manuscript is well-written and easy to follow. The following comments must be addressed before consideration for publication.

Major comments:

1. I recommend some of the figures can be moved from the main text to supplement, such as Fig. 7 and Fig. 8. Figures 3 and 4 can be combined into one figure. Figures 10 and 11 can be combined also. I prefer a compact and relatively short manuscript to introduce such a new instrument.

2. In Sect. 3.1.2, the double charged aerosols effect of the calibration of $F_{act} - S_{aerosol}$ relation was discussed. Did you consider the double charged effect in ambient measurement?

3. In Sect. 3.1.3, the water depletion in the activation tube by high number concentration was discussed. I was wondering do you pre-humidify the particle for ambient aerosol measurement. What is the total particle number concentration of the ambient measurement in this study? In Lines 217-218, it was mentioned that the aerosol concentration needs to be considered when it is high. Do you think is it worth testing the water depletion by high number concentration (such as $>1000$ cm$^{-3}$) with lab-generated particles or in a polluted environment?

4. In Sect. 4.1, the AS and Su mixed particles are internally or externally mixed? If externally mixed, can it be seen in the $F_{act} - S_{aerosol}$ curves?

Monir comments:

Line 50-60: When talking about the fast measurement of size-resolved $\kappa$, Zhang et al. (2021) introduced a novel method/technique to rapidly measure the size-resolved $\kappa$ values under subsaturation.

Line 79: I realized that Fig. S1 was originally from Su et al. (2016). You probably need to mention that in the manuscript. I recommend moving Fig. S1 from supplement to main text because this figure is important for readers to understand the $F_{act}$.

Line 114: Why particle number concentration is the number concentration of condensation nuclei? $N_{CCN}$ and $N_{CN}$ are the same abbreviations?

**References:**

Zhang, J., Spielman, S., Wang, Y., Zheng, G., Gong, X., Hering, S., and Wang, J.: Rapid measurement of RH-dependent aerosol hygroscopic growth using a humidity-controlled fast integrated mobility spectrometer (HFIMS), Atmos. Meas. Tech. Discuss. [preprint], https://doi.org/10.5194/amt-2021-90, in review, 2021.

---

## Author Comment (AC1)

**Response to Reviewer #2**

*This manuscript presents a novel broad supersaturation scanning CCN (SB2-CCN) system, which can measure the CCN activity with a high time resolution. Overall, this manuscript clearly explained the set-up and calibration of the instrument, and applied it in the field measurement. This manuscript is well-written and easy to follow. The following comments must be addressed before consideration for publication.*

A: We thank the reviewer for encouraging and helpful comments on our manuscript. We believe that the quality of our manuscript is improved as we reflect the reviewer's comments. Below each of the questions/comments is written first with the Italic font, and then our response is followed with the normal font. We marked newly added or revised sentences by highlighting them in the revised manuscript.

**Comments**

*Q1: I recommend some of the figures can be moved from the main text to supplement, such as Fig. 7 and Fig. 8. Figures 3 and 4 can be combined into one figure. Figures 10 and 11 can be combined also. I prefer a compact and relatively short manuscript to introduce such a new instrument.*

A: I combined Fig.3 and 4 (now 'Figure.4'), and Fig. 10 and 11 (now 'Figure.9') as you suggested. And, Figure 8 (now 'Figure S4') is moved to the supplement. Figure 8 is still in the main text to show the curve fitting procedure.

*Q2: In Sect. 3.1.2, the double charged aerosols effect of the calibration of Fact – Saerosol relation was discussed. Did you consider the double charged effect in ambient measurement?*

A: For ambient measurement in this study, we do not consider the double charged effect for ambient measurement. We added the sentence "The double charged effect is not considered in the inter-comparison experiment." (Line 293-294). We will discuss the correction of multiply charged particles in detail along with the field campaign results in the next paper. Instead, in section 3.1.2, we performed an additional calculation for ideal activation fraction with atmospheric relevant particle size distribution. We add the figure in Fig.S4 and add sentence. "Assuming the atmospheric relevant particle number size distribution with $N = 1000\ cm^{-3}, D_g = 80\ nm, and\ \sigma_g = 1.5$ from Rose et al. (2011), $F_{act\_double}$ is up to 0.05 (Fig. S4). It is noted that aerosols with $\kappa$ of 0.3 is assumed to be internally mixed, and $S_{max}$ is set to be 0.2%. Although the effect of doubly charged particle in Fig. S4 is not significant, the effect of doubly charged particle cannot be ignored if $D_g$ or $\sigma_g$ becomes large in specific environments or conditions." (Line 200-204).

*Q3: In Sect. 3.1.3, the water depletion in the activation tube by high number concentration was discussed. I was wondering do you pre-humidify the particle for ambient aerosol measurement. What*

*is the total particle number concentration of the ambient measurement in this study? In Lines 217-218, it was mentioned that the aerosol concentration needs to be considered when it is high. Do you think is it worth testing the water depletion by high number concentration (such as >1000 cm-3) with lab-generated particles or in a polluted environment?*

A: Pre-humidifier system is not included during the ambient aerosol measurement in this study. Particle number concentration at each selected diameter during the measurement ranged from 0 to 74 $cm^{-3}$ and average number concentration was 18 $cm^{-3}$. In other words, pre-humidifier system is not necessary for this measurement. We also performed the test of water depletion by high number concentration (Maximum $N_{CN} \sim 1100$ $cm^{-3}$). Slope was similar to the experiment in the main manuscript. For example, $F_{act}$ decreases 1.3 % (120 nm) per increase of 100 $cm^{-3}$ in the number of particles (Figure R1). As BS2-CCN measurement system is based on the size-resolved measurement, we expect that particle number concentration of selected dry diameter will be within the range of concentration presented in the manuscript for the typical observations of ambient aerosols except for special high concentration cases (Figure R2).

[Figure]

**Figure R 1: Average and standard deviation (error bar) of $F_{act}$ depending on the number concentration $N_{CN}$ for 60 nm (black), 80 nm (red) and 120 nm (blue) of ammonium sulfate under the dT=7.7 K condition**.

[Figure]

**Figure R 2: Example of particle number size distribution data measured from SMPS in urban area during new particle formation event. Particle number concentrations at mode diameter are about 950 cm-3.**

For calibration experiment with lab-generated conditions, water depletion without pre-humidifier system can affect the calibration curve if particle number concentration is too high. The slope between particle number concentration and $F_{act}$ might be slightly different depending on the performance of CCNC itself. Therefore, we mention about recommended number concentration for compact instrumental setup without the pre-humidifier in the manuscript and the necessity of pre-humidifier system for special cases in the manuscript "In other words, a compact instrumental setup without the pre-humidifier system is sufficient for the BS2-CCNC calibration experiment as well as the measurement if aerosol particles are kept below ~ $3 \times 10^2$ cm-3. Otherwise, we need a pre-humidifier system for high aerosol number concentration condition to avoid the decrease of $F_{act}$." (Line 225-227).

*Q4: In Sect. 4.1, the AS and Su mixed particles are internally or externally mixed? If externally mixed, can it be seen in the Fact – Saerosol curves?*

A: AS and Su mixed particle are internally mixed as we generated these aerosols by the solution of their mixtures. I follow the method that dissolving each component into the pure water to form dilute solution and generate atomized aerosol to generate internally mixed aerosols (Shi et al. 2012; Jing et al. 2016). Also, we get a smooth activation curve (no plateau in the middle of the curve, implying internally mixed) from DMA-CCN measurement. To be clear, I revise the sentence in the manuscript "We used an atomizer to generate internally-mixed nanoparticles with diameters of 30 – 160 nm by spraying a mixed solution of succinic acid and ammonium sulfate, which each pure component was completely dissolved in pure water obtained from a Mili-Q water purification system (Line 274-276).

$F_{act} - S_{aerosol}$ curve can be obtained from pure component with known aerosols as $S_{aerosol}$ is calculated based on κ-Köhler theory. For externally mixed aerosols, Su et al. (2016) suggest combining BS2-CCNC with complementary measurement (e.g. DMT-CCNC) to resolved the multiple κ mode by decoupling the mixed information and extracting the signal of each mode. If assuming aerosols with two log-normally distributed κ mode (κ of 0.3 and 0.01) with the same $\sigma_\kappa$ (geometric standard deviation of κ), lower κ values are retrieved from model simulation with a BS2 approach itself ("apparent" in Fig.R3b) than those calculated with complementary measurement of the DMT-CCNC ("corrected" in

Fig.R3a). Below are equations, Eq. (R1) – (R3), used for calculation based on the assumptions, and the results are shown in the Fig.R3.

$$h(\kappa) = \frac{a_{\kappa 0.01}}{\sqrt{2\pi}log\sigma_\kappa}exp(-\frac{(log\kappa - log0.01)^2}{2(log\sigma_\kappa)^2}) + \frac{a_{\kappa 0.3}}{\sqrt{2\pi}log\sigma_\kappa}exp(-\frac{(log\kappa - log0.3)^2}{2(log\sigma_\kappa)^2}) \tag{R1}$$

$$a_{\kappa 0.01} = 0.3 \times log_{10}(D_d/30) \;and\; a_{\kappa 0.01} = 1 - a_{\kappa 0.3} \tag{R2}$$

$$F_{act} = a_{\kappa 0.3} \times F_{act,\kappa\,0.3} + a_{\kappa 0.01} \times F_{act,\kappa\,0.01} \tag{R3}$$

Where, $h(\kappa)$ is the fractional probability distribution function of hygroscopicity (Su et al. 2010), $a_{\kappa 0.01}$ and $a_{\kappa 0.3}$ present the number fraction of the two modes, and $F_{act}$ represents an average of individual mode weighted by their number fraction

[Figure]

**Figure R 3. (a) Supersaturation required to activate the less hygroscopic mode by the DMT-CCNC. The shaded isolines show the normalized size-resolved hygroscopicity distribution $h(\kappa)$, the probability density function of a two mode κ distribution (Rose et al. 2011). The colored isolines represent the supersaturation required to activate particles for a certain dry diameter $D_d$ and κ. (b) Model simulation of κ retrieved for the more hygroscopic mode by BS-CCNC with (labeled as "corrected") and without (labeled as "apparent) complementary measurement of the DMT-CCNC. Reprinted from Su et al. (2016) under the Creative Commons Attribution 4.0 License.**

'

*Q5: Line 50-60: When talking about the fast measurement of size-resolved κ, Zhang et al. (2021) introduced a novel method/technique to rapidly measure the size-resolved κ values under sub-saturation.*

*Zhang, J., Spielman, S., Wang, Y., Zheng, G., Gong, X., Hering, S., and Wang, J.: Rapid measurement of RH-dependent aerosol hygroscopic growth using a humidity-controlled fast integrated mobility*

*spectrometer (HFIMS), Atmos. Meas. Tech., 14, 5625–5635, https://doi.org/10.5194/amt-14-5625-2021, 2021.*

A: We added the sentence "Zhang et al. (2021) introduce a novel measurement technique using a humidity-controlled fast integrated mobility spectrometer (HFIMS) to measure the size-resolved κ rapidly under the sub-saturated condition." (Line 60-62) as suggested.

*Q6: Line 79: I realized that Fig. S1 was originally from Su et al. (2016). You probably need to mention that in the manuscript. I recommend moving Fig. S1 from supplement to main text because this figure is important for readers to understand the Fact.*

A: Following the reviewer's suggestion, we have moved Fig. S1 to the main text. Besides in the figure caption, we also include another reference in the main text "Fig.1, Reprinted from Su et al., (2016) under the Creative Commons Attribution 4.0 License.".

*Q7: Line 114: Why particle number concentration is the number concentration of condensation nuclei? NCCN and NCN are the same abbreviations?*

A: $N_{CCN}$ and $N_{CN}$ are not the same abbreviation, number concentration of condensation nuclei for $N_{CN}$ and number concentration of cloud condensation nuclei for $N_{CCN}$. Condensation nuclei are tiny particles in the air on which water vapor condenses and the term '$N_{CN}$' is commonly used to describe the total particle number concentration (Rose et al. 2008; Pöhlker et al. 2018; Gao et al. 2020) and can be measured by condensation particle counter (CPC).

**References:**

Gao, Y., Zhang, D., Wang, J., Gao, H., and Yao, X.: Variations in $N_{cn}$ and $N_{ccn}$ over marginal seas in China related to marine traffic emissions, new particle formation and aerosol aging, Atmos. Chem. Phys., 20, 9665–9677, https://doi.org/10.5194/acp-20-9665-2020, 2020.

Jing, B., Tong, S., Liu, Q., Li, K., Wang, W., Zhang, Y., and Ge, M.: Hygroscopic behavior of multicomponent organic aerosols and their internal mixtures with ammonium sulfate, Atmos. Chem. Phys., 16, 4101–4118, https://doi.org/10.5194/acp-16-4101-2016, 2016.

Pöhlker, M. L., Ditas, F., Saturno, J., Klimach, T., Hrabě de Angelis, I., Araùjo, A. C., Brito, J., Carbone, S., Cheng, Y., Chi, X., Ditz, R., Gunthe, S. S., Holanda, B. A., Kandler, K., Kesselmeier, J., Könemann, T., Krüger, O. O., Lavrič, J. V., Martin, S. T., Mikhailov, E., Moran-Zuloaga, D., Rizzo, L. V., Rose, D., Su, H., Thalman, R., Walter, D., Wang, J., Wolff, S., Barbosa, H. M. J., Artaxo, P., Andreae, M. O., Pöschl, U., and Pöhlker, C.: Long-term observations of cloud condensation nuclei over the Amazon rain forest – Part 2: Variability and characteristics of biomass burning, long-range transport, and pristine rain forest aerosols, Atmos. Chem. Phys., 18, 10289–10331, https://doi.org/10.5194/acp-18-10289-2018, 2018.

Rose, D., Gunthe, S. S., Mikhailov, E., Frank, G. P., Dusek, U., Andreae, M. O., and Pöschl, U.: Calibration and measurement uncertainties of a continuous-flow cloud condensation nuclei counter (DMT-CCNC): CCN activation of ammonium sulfate and sodium chloride aerosol particles in theory and experiment, Atmos. Chem. Phys., 8, 1153–1179, https://doi.org/10.5194/acp-8-1153-2008, 2008.

Rose, D., Gunthe, S. S., Su, H., Garland, R. M., Yang, H., Berghof, M., Cheng, Y. F., Wehner, B., Achtert, P., Nowak, A., Wiedensohler, A., Takegawa, N., Kondo, Y., Hu, M., Zhang, Y., Andreae, M. O., and Pöschl, U.: Cloud condensation nuclei in polluted air and biomass burning smoke near the mega-city Guangzhou, China –

Part 2: Size-resolved aerosol chemical composition, diurnal cycles, and externally mixed weakly CCN-active soot particles, Atmos. Chem. Phys., 11, 2817–2836, doi:10.5194/acp-11- 2817-2011, 2011.

Shi, Y. J., Ge, M. F., and Wang, W. G.: Hygroscopicity of internally mixed aerosol particles containing benzoic acid and inorganic salts, Atmos. Environ., 60, 9–17, doi:10.1016/j.atmosenv.2012.06.034, 2012

Su, H., Rose, D., Cheng, Y. F., Gunthe, S. S., Massling, A., Stock, M., Wiedensohler, A., Andreae, M. O., and Pöschl, U.: Hygroscopicity distribution concept for measurement data analysis and modeling of aerosol particle mixing state with regard to hygroscopic growth and CCN activation, Atmos. Chem. Phys., 10, 7489–7503, https://doi.org/10.5194/acp-10-7489-2010, 2010

---

## Author Comment (AC2)

**Response to Reviewer #1**

*Kim et al. present the calibration of the broad supersaturation scanning CCN instrument. The instrument is a modified version of the DMT CCN, where the aerosol is entering the growth tube in a spatially distributed manner. As a result, particles experience a range of supersaturation. When fed with monodisperse particles, the activated fraction can be related directly to the hygroscopicity parameter of the aerosol at that size. The manuscript presents experimental calibration data for this instrument. Comparison with regular size-resolved CCN measurements shows reasonable agreement in the derived kappa for the two methods.*

*Overall the paper is well written. The method is clever and promises a faster time-response measurement with relatively small modifications to an existing widely available commercial instrument. The manuscript is relevant to the readers of AMT and I recommend publication if the following comments can be addressed.*

A: We thank the reviewer for helpful comments on our manuscript. We expect our manuscript to improve more qualitatively when your comments are reflected. Below each of questions/comments is written first with the Italic font, and then our response is followed with the normal font. We marked newly added or revised sentences by highlighting them in the revised manuscript.

**Major comments.**

*Q1: Influence of multiply charged particles: The authors select the most optimistic scenario to conclude that the effect of multiply charged particles is small. Conditions of lower supersaturation and larger mode diameter will have a much more significant influence of multiply charged particles. Figure 4 should include s = 0.1% and Dg = 150 nm to bound the magnitude of the effect, which may be especially relevant in some ambient cases scenarios.*

A: As you suggested, we calculated ideal activation fraction with assumed atmospheric-relevant log-normal particle size distribution. As we think 150 nm of $D_g$ is too large and $D_g$ of many cases is less than 100nm, we adopted particle number size distribution from Rose et al. (2011) with $N = 1000\ cm^{-3}, D_g = 80\ nm, and\ \sigma_g = 1.5$. As the BS2-CCNC uses the $S_{tube}$ distribution in the chamber, not a single supersaturation, the supersaturation is too low in the edge of chamber if we set 0.1% for $S_{max}$. Therefore, we set 0.2% for $S_{max}$, instead of 0.1. Additionally, we assumed aerosols with κ of 0.3 are internally mixed. According to the Fig. R1, $F_{act\_double}$ is up to 0.05 implying that the effect of doubly charged particle is not that significant but we still need to consider the effect of doubly charged particle for ambient aerosol measurement if $D_g$ or $\sigma_g$ becomes large. As Section 3 is focused on the calibration experiment, we added this result in Supplement (Figure S4) and sentence "Assuming the atmospheric relevant particle number size distribution with $N = 1000\ cm^{-3}, D_g = 80\ nm, and\ \sigma_g = 1.5$ from Rose et al. (2011), $F_{act\_double}$ is up to 0.05 (Fig. S4). It is noted that aerosols with κ of 0.3 is assumed to be internally mixed, and $S_{max}$ is set to be 0.2%. Although the effect of doubly charged particle in Fig. S4 is not significant, the effect of doubly charged particle cannot be ignored if $D_g$ or $\sigma_g$ becomes large in specific environments or conditions." (Line 200-204).

[Figure]

**Figure R 1: Calculated ideal activation fraction for log-normally distributed, charge-equilibrated particles transmitted BS2-CCNC system Shown are assumed particle size distribution (black-solid line, left ordinate, $N = 1000 \, cm^{-3}, D_g = 80 \, nm, and \, \sigma_g = 1.5$, total activation fraction (red solid line), activation fractions by singly charged particle (red dashed line) and doubly charged particle (blue dashed line). It is noted that $S_{max}$ is set to be 0.2% in this calculation.**

*Q2: Time resolution: The promise of the technique is that kappa can be measured at much higher time resolution. However, the manuscript does not really show this very well.*

*Figure 11 shows temporally averaged data. Perhaps this is necessary because the 1 min data are too noisy? If that is the case, it would undercut the argument of improved time resolution.*

*Related, there is a concern on what went into the average. Technically, the comparison should be for the size closest to the activation diameter of the size resolved CCN, which changes with time. The authors should be more precise when matching the kappa values in the comparison (i.e. only include +/- 1 size bin in their kappa inter-comparison).*

*In general, the authors should discuss time resolution in a more nuanced manner. When pressed, scanning flow CCN and scanning mobility CCN can find an activation spectrum in 30s to 1 min time, which would provide a kappa value every minute. The D50/supersaturation could be adjusted by changing the flow rate after each scan. This may be inferior to finding the kappa value at a fixed size which the BS2 technique does. However, at face value the time resolution of what can be achieved with traditional methods would seem much more similar to what is achieved in this work, although that setup might be easily improved by changing the configuration (see below). Related, it would be good to add discussion on the minimum time needed to get a kappa measurement. Why was 1 min selected? What determines the quality of the measurement? Is it the number of counts? If so can this value be specified? In principle, it would seem possible to run the system with a continuously scanning DMA (e.g. a 3 min SMPS scan) and then report the kappa data in a few discrete size bins. The feasibility of such an approach would depend on the time required to obtain a good kappa characterization. The authors should provide detailed comments on what may or may not be possible with this technique.*

A: For Fig.11, we use 1 min averaged data to not only compare the κ value between DMA-CCN and BS2-CCN measurement but also obtain more reliable result from both measurements. Instrumental setup for inter-comparison experiment is on Fig.S6. It doesn't mean that 1 minute is the minimum time resolution to obtain κ from BS2-CCN system.

For detailed analysis of time resolution of BS2-CCN system, we added a figure of exemplary $D_p$ scan with ammonium sulfate ($D_p$: 20 – 100 nm, dT= 10 K). We change diameter every 40 seconds and plot each of 1 second data (Figure. R2, Figure S2 in the supplement). For scanning, it takes about 10 seconds (maximum) to stabilize immediately after changing the particle size. We then averaged the 30-second data and display the average and standard deviation value of each dry diameter in orange together in Fig. R2a. The details of average and standard deviation value of each dry diameter are presented in Table R1. Additionally, we calculated the absolute deviation value for each data. The absolute deviations increased during the stabilization especially when CCN number concentration starts to increase. After stabilization, absolute deviation was mostly less than 0.05 except when $F_{act}$ value is higher than 0.85. It is noted that we exclude the data which $F_{act}$ is higher than 0.85 for analysis (Line 295 – 296). In other words, we could get reliable κ value in 1 second time resolution after stabilization as κ value is derived from $F_{act}$ directly. We also presented κ distribution which corresponds to the $F_{act}$ value of particle ranged from 50 nm to 150 nm in Fig.S4. Although we need to consider the stabilization time (~10s) after changing particle size for scanning measurement, we could derive κ value in 1 second if we set a single particle size implying that it is applicable to aircraft measurement which requires high-time resolution. Also, BS2-CCN system minimize the potential problem of aerosol volatilization and technical complexity as the BS2-CCN system uses constant temperature gradient and flow rate.

We add Fig. R2 in supplement information (Figure S2 in the Supplement) and sentences "Figure S2 is an exemplary $D_p$ scan with ammonium sulfate particle to examine the time resolution of BS2-CCN system. $F_{act}$ of each diameter is measured every 40 seconds including stabilization. It is noted that $N_{CN}$ and $N_{CCN}$ data has 1s time resolution and thereby $F_{act}$ data with 1s time resolution are initially available. For scanning, it takes up to 10 seconds to stabilize immediately after changing the particle size. Absolute deviation of $F_{act}$ is mostly less than 0.05 except when $F_{act}$ is higher than 0.85. In other words, we could get reliable κ value, derived by $F_{act}$ directly, in 1 second time resolution after stabilization for $D_p$ scan measurement. Additionally, if we set a single particle size, we could derive κ value in 1 second time resolution. However, in this study, for calibration experiment, we use 1-min average data including stabilization time to calculate $F_{act}$ value corresponding to each $D_d$." (Line 119 – 126).

For inter-comparison experiment we set 19 dry diameters of 40 – 250 nm and select kappa value of close diameter in BS2-CCN measurement to the critical diameter obtained from DMA-CCN measurement. We mentioned this in Line 304 – 305 "Therefore, as shown in Fig. 9c, we used κ value of BS2-CCN measurement by selecting the particle diameter close to the average $D_c$ of the DMA-CCN measurement for the inter-comparison."

**Table R 1: Average and standard deviation value of $F_{act}$ for each dry diameter.**

| Dp [nm] | Average $F_{act}$ | Standard deviation of $F_{act}$ |
|---|---|---|
| 20 | 0.00316866 | 0.00218781 |
| 25 | 0.00785123 | 0.00269424 |
| 30 | 0.03612815 | 0.00484231 |
| 35 | 0.05542066 | 0.00519961 |

| 40 | 0.26255714 | 0.01788727 |
| 45 | 0.47451548 | 0.0196648 |
| 50 | 0.61359299 | 0.01808588 |
| 55 | 0.71061904 | 0.0278038 |
| 60 | 0.76992656 | 0.02853376 |
| 65 | 0.80506852 | 0.02585054 |
| 70 | 0.84679062 | 0.0303469 |
| 75 | 0.88028529 | 0.02867314 |
| 80 | 0.91009636 | 0.02875085 |
| 85 | 0.91778563 | 0.03408703 |
| 90 | 0.93236908 | 0.03541124 |
| 95 | 0.95317282 | 0.03227177 |
| 100 | 0.94493573 | 0.04476325 |

[Figure]

**Figure R 2: Exemplary $D_p$ scan (20-100 nm) of lab generated ammonium sulfate. (a) 1 second data of activated fraction ($F_{act}$), marked in black dot (left ordinate), particle diameter (red line, right ordinate). Average and standard deviation of each diameter (30-second average data except for 10 seconds of stabilization) are presented in orange square with bar. (b) Absolute deviation of $F_{act}$ (black dot, left ordinate) and particle diameter (red line, right ordinate): The grey shaded box indicates the stabilization time (~10 seconds) of each particle diameter. (c) Particle number concentration ($N_{CN}$, red dot) and CCN number concentration ($N_{CCN}$, blue dot). Time resolution of each data point is 1 second and the particle diameter is changed every 40 seconds. $S_{max}$ is set to be 10 K (0.8%).**

*Q3: The authors should discuss the new inlet in more detail. Were CFD simulations used to make sure that the flow is laminar? What are the limits of the angle to achieve laminar low? For maximum impact, the authors should consider publish their CAD drawings under a non commercial use license so that others can more easily implement this technique. (Publication of the CAD drawing is not a requirement for publication in AMT, though in the reviewers opinion it should be).*

A: As the reviewer suggested, we conducted computational fluid dynamics (CFD) simulation with COMSOL Multiphysics (version 5.6). With our inlet design, laminar flow can be achieved regardless of the angle due to the low velocity. However, if the angle becomes too large (Figure R3b), flow separation occurs at the point where the aerosol and sheath flow meet, and there is a section where constant steady rotation of flow occurs. It means that the laminar flow cannot be achieved if there is no sheath flow, only aerosol flow. Everything above 15mm of cone length (65mm of the original design) doesn't show a separation of flowlines from the wall. We add the description of flow with the newly designed inlet in the main manuscript "According to the computational dynamic simulation result (COMSOL Multiphysics, version 5.6) of flow streamline and the relative particle concentration in Fig. A2, laminar flow inside the activation tube can be achieved with our new inlet design. Additionally, this new inlet allows for maintaining stable low sheath-to-aerosol flow ratios (SAR), for which monotonic $F_{act} - S_{aerosol}$ relation can then be obtained." (Line 102-105) and add the figure of flow streamline and concentration (Figure R3a only) to support the description in Appendix A (Figure. A2).

Unfortunately, CAD drawing cannot be added in the manuscript. We showed the detailed front perspective view and Longitudinal sectional view of an embodiment of the newly designed inlet and the explanation of each part in Appendix A and Fig. A1. Compared to other instrument papers, including Fig.2 of Robert and Nenes (2005) for cloud condensation nuclei counter (CCNC), Jayne et al. (2000) for aerosol mass spectrometer (AMS), Fig.1 and Fig.2 of Pinterich et al. (2017) for a humidity-controlled fast integrated mobility spectrometer (HFIMS), we believe that our manuscript provides sufficient information on the concept and description of new measurement system as well as the design of new inlet. More detailed information of the newly designed inlet will be provided by the corresponding author (Dr. Hang Su, h.su@mpic.de) upon request and we are always welcome to collaborate.

[Figure]

**Figure R 3.Results of computational fluid dynamics simulation with (a) original inlet design and (b) modified inlet design. A solid black line and the color bar indicate the flow streamline in the velocity field and the relative particle concentration [mol/m³], respectively. It is noted that the figure presents the half side of a longitudinal sectional view of Fig.A1(b) and the x and y axes represent the length of the inlet (units are meters). The aerosol and sheath flow go from the bottom (-y) to the top (+y).**

*Q4: "Data can be accessed by contacting the corresponding author. ": This is incompatible with the data policy of AMT.*

A: Following the data policy of AMT, we integrated the data of figures in the manuscript and uploaded the data on the Edmond, the open research data repository of the Max Planck Society. And, we add link for data and sentences "Data can be downloaded from Edmond, open research data repository of Max Planck Society (https://edmond.mpdl.mpg.de/imeji/collection/pohD2XdTlrMwzka7), and raw data are available upon request from the corresponding author (h.su@mpic.de)" in Data Availability.

**Other comments**

*Q5: Figure 3: The distribution doesn't peak at 50 nm as stated in the text.*

A: The peak diameter of the distribution on Figure 3 (now Figure 4) is 50 nm. Due to the x-axis, it is not clear to show the peak diameter. Therefore, we changed the x-axis of Fig.4 (a) and (b). Also, we had small mistake in caption. We changed value of $\sigma_g$ from 1.4 to 1.5.

**Reference**

J.T. Jayne, D.C. Leard, X. Zhang, P. Davidovits, K.A. Smith, C.E. Kolb, and D.R. Worsnop.: Development of an aerosol mass spectrometer for size and composition analysis of submicron particles, Aerosol Sci. Technol., 33, 49-70, 2000.

Pinterich, T., Spielman, S. R., Wang, Y., Hering, S. V., and Wang, J.: A humidity-controlled fast integrated mobility spectrometer (HFIMS) for rapid measurements of particle hygroscopic growth, Atmos. Meas. Tech., 10, 4915–4925, https://doi.org/10.5194/amt-10-4915-2017, 2017.

Roberts, G. C. and Nenes, A.: A Continuous-Flow Streamwise Thermal-Gradient CCN Chamber for Atmospheric Measurements', Aerosol Science and Technology, 39:3, 206 – 221, https://doi.org/10.1080/027868290913988, 2005.

Rose, D., Gunthe, S. S., Su, H., Garland, R. M., Yang, H., Berghof, M., Cheng, Y. F., Wehner, B., Achtert, P., Nowak, A., Wiedensohler, A., Takegawa, N., Kondo, Y., Hu, M., Zhang, Y., Andreae, M. O., and Pöschl, U.: Cloud condensation nuclei in polluted air and biomass burning smoke near the mega-city Guangzhou, China – Part 2: Size-resolved aerosol chemical composition, diurnal cycles, and externally mixed weakly CCN-active soot particles, Atmos. Chem. Phys., 11, 2817–2836, doi:10.5194/acp-11- 2817-2011, 2011.